# Assessing the dynamics of soil salinity with time-lapse inversion of electromagnetic data guided by hydrological modelling

Mohammad Farzamian[1,2], Dario Autovino[3], Angelo Basile[3], Roberto De Mascellis[3], Giovanna Dragonetti[4], Fernando Monteiro Santos[2], Andrew Binley[5], Antonio Coppola[6]

[1]Instituto Nacional de Investigação Agrária e Veterinária, Oeiras, 2780-157, Portugal
[2]Instituto Dom Luiz, Faculdade de Ciências da Universidade de Lisboa, Lisboa, 1749-016, Portugal
[3]Institute for Mediterranean Agricultural and Forestry Systems, National Research Council, Portici (NA), 80055, Italy
[4]Mediterranean Agronomic Institute of Bari, Valenzano (BA), 70010, Italy
[5]Lancaster Environment Centre, Lancaster University, Lancaster, LA1 4YQ, United Kingdom
[6]School of Agricultural, Forestry, Food and Environmental Sciences, University of Basilicata, Potenza, 85100, Italy

*Correspondence to*: Angelo Basile (angelo.basile@cnr.it)

**Abstract.** Irrigated agriculture is threatened by soil salinity in numerous arid and semiarid areas of the world, chiefly caused by the use of highly salinity irrigation water, compounded by excessive evapotranspiration. Given this threat, efficient field assessment methods are needed to monitor the dynamics of soil salinity in salt-affected irrigated lands and evaluate the performance of management strategies. In this study, we report on the results of an irrigation experiment with the main objective of evaluating time-lapse inversion of electromagnetic induction (EMI) data and hydrological modelling in field assessment of soil salinity dynamics. Four experimental plots were established and irrigated twelve times during a two-month period, with water at four different salinity levels (1, 4, 8 and 12 $dSm^{-1}$) using a drip irrigation system. Time-lapse apparent electrical conductivity ($\sigma_a$) data were collected four times during the experiment period using a CMD Mini-Explorer. Prior to inversion of time-lapse $\sigma_a$ data, a numerical experiment was performed by 2D simulations of the water and solute infiltration and redistribution process in synthetic transects, generated by using the statistical distribution of the hydraulic properties in the study area. These simulations gave known spatio-temporal distribution of water contents and solute concentrations, and thus of bulk electrical conductivity ($\sigma_b$), which in turn were used to obtain known structures of apparent electrical conductivity, $\sigma_a$. These synthetic distributions were used for preliminary understanding of how the physical context may influence the EMI-based $\sigma_a$ readings carried out in the monitored transects, as well as to optimize the smoothing parameter to be used in the inversion of $\sigma_a$ readings. With this prior information at hand, we inverted the time-lapse field $\sigma_a$ data and interpreted the results in terms of concentration distributions over time. The proposed approach, using preliminary hydrological simulations to understand the potential role of the variability of the physical system to be monitored by EMI, may actually allow a better choice of the inversion parameters and interpretation of EMI readings, thus increasing the potentiality of using the electromagnetic induction technique for rapid and non-invasive investigation of spatio-temporal variability in soil salinity over large areas.

## 1. Introduction

Soil salinization may be of a primary nature, when salt accumulation arises through pedogenetic processes, or of secondary origin, due either to abiotic factors such as excessive evaporation or sea-water infiltration, or resulting from human intervention, chiefly use of saline water irrigation (Geeson et al., 2002). Approximately 20% of irrigated land (45 million ha) that produces one-third of the world's food, is salt-affected (Shirvastava and Kumar, 2015) and it is estimated that soil salinity affects 1 million hectares in the European Union, mainly in the Mediterranean countries (Toth et al., 2008).

Effective agricultural management in many areas relies on a good understanding of the effects of irrigation on the spatial and temporal variability of soil salinity (Coppola et al., 2015). However, it is very difficult to assess soil salinity on a management scale using traditional methods. Soil salinity is traditionally assessed by measuring the electrical conductivity of a saturated soil paste ($EC_e$) in the laboratory; however, this technique is labour intensive, time-consuming and costly, given the large number of soil samples that need to be collected. Alternatively, Time Domain Reflectometry (TDR), a non-destructive electromagnetic technique, can be used in the field for simultaneous measurements of water content, $\theta$, and bulk electrical conductivity, $\sigma_b$ (Coppola et al., 2011b). While the TDR method can provide accurate information from the local measurements, the measurement support volume of sensors is limited to a few centimetres, thus extension of the information to a large area can be problematic (Coppola et al., 2016; Gonçalves et al., 2017, Dragonetti et al., 2018).

The electromagnetic induction (EMI) technique is widely used as an alternative to traditional techniques for soil salinity assessment. It allows rapid non-invasive, reliable and repeatable measurements at a smaller cost than traditional methods. This technique measures the soil electrical conductivity which is primarily a function of soil salinity, soil texture, moisture content, and cation exchange capacity; however, in a saline soil, the salinity is generally the dominant factor responsible for the spatio-temporal variability of soil electrical conductivity (Corwin and Lesch, 2005).

In the last few decades, EMI techniques have been used increasingly to estimate soil salinity from apparent electrical conductivity ($\sigma_a$) measurements (Lesch et al., 1995; Triantafilis et al., 2000; Corwin et al., 2006; Ganjegunte et al., 2014). $\sigma_a$ is the weighted average of the soil electrical conductivity distribution in the soil volume. In order to obtain the depth-distribution of $\sigma_b$ from $\sigma_a$, a site-specific empirical calibration between $\sigma_a$ and soil salinity measured at different depths can be applied by different approaches such as multiple regression (Triantafilis et al., 2000; Amezketa, 2006; Yao and Yang, 2010; Coppola et al., 2016), modelled coefficients (Slavich and Petterson, 1990), theoretical coefficients calculated with theoretical EMI depth response functions (Cook and Walker, 1992) or empirical-mathematical coefficients (Corwin and Rhoades, 1984).

Alternatively, to assess the distribution of $\sigma_b$ with depth, $\sigma_a$ data collected in the field can be modelled through an inversion process. Several inversion methods have been proposed to obtain the $\sigma_b$ from the measured $\sigma_a$ data, including the gradient-based inversion technique (Monteiro Santos, 2004; von Hebel et al., 2014; Schamper et al., 2012; Farzamian et al., 2015a) and probabilistic inversion (Jadoon et al., 2017; Moghadas, 2019; Shanahan et al., 2015). Recently, multi-coil EMI measurements and inversion algorithms have been widely used for mapping soil salinity and sodicity distribution in quasi 2D

(e.g. Goff et al., 2014; Farzamian et al., 2019; Paz et al., 2019, 2020a) and 3D (e.g. Huang et al., 2017). However, the potential of this method in assessing temporal variability of soil salinity has not been fully explored. Several time-lapse inversion methods for direct-current resistivity methods have been developed. These include the ratio method (Daily et al., 1992), the difference inversion (LaBrecque and Yang, 2001) and more recently, 4-D space–time algorithms (Kim et al., 2009). A number of studies have also demonstrated how the use of time-lapse inversion algorithms can reduce the inversion

artefacts (e.g. Hayley et al., 2011) and improve the quantitative investigations of geophysical monitoring (e.g. Farzamian et al., 2015b). However, the usefulness of the time-lapse inversion algorithms in modelling EMI data has not been attempted yet to assess soil salinity dynamic and only few studies have been conducted to estimate soil water content changes (Huang et al., 2016; Whalley et al., 2017). Besides, a prerequisite for such an approach concerns the reliability of the inversion of the EMI result. In fact, inverting profile-integrated EMI data to obtain the vertical distribution of $\sigma_b$ is an ill-posed problem,

suffering from non-uniqueness (the problem has more than one solution) and instability (incomplete data and measurement errors can lead to large changes in the parameters (e.g. Tarantola, 1987). Ill-posedness is generally treated by regularizing the inverse solution. However, different regularization schemes and parameters can have a significant impact on the results (e.g. Dragonetti et al., 2018; Zare et al., 2020), thus, inversion results of EMI data are always affected by uncertainties, which can be minimized in case of prior information from the experiment.

In this direction, preliminary numerical simulations of the same hydrological processes to be monitored by an EMI sensor, by applying real boundary conditions measured during an EMI sensor monitoring campaign, may be especially helpful to figure out the response to be expected by the sensor and its variability in the space and time, and may allow for a more rational choice of the EMI inversion parameterization. In other words, hydrological simulations may help provide an "a priori" knowledge of "where the EMI inversion has to go". In any case, this would require the actual distribution of the

hydraulic properties along the transect or, more in general, in the field to be monitored by the EMI. One can immediately realize that this is quite utopian, especially when the area to be monitored is relatively large (as previously recalled in the case of EMI measurements). By contrast, it is more common that, for the study area, one has available the statistical distribution of hydraulic properties. Thus, the statistical distribution of the hydraulic properties may be used for generating synthetic (but realistic in a probabilistic view) equiprobable realizations of the physical conditions the EMI sensor will

potentially experience during the monitoring, which may be used for addressing the inversion of EMI reading.
The main objective of this paper is to propose an approach to improve the parameters optimization and the constrains in time-lapse EMI inversion using soil water and solute modelling. In the paper we will show how the synthetic tests may be used to guide the optimization of inversion parameters and understanding of the impact of solute concentration and water content variations on EMI $\sigma_a$ readings.

The key features of this study are: i) performing a controlled irrigation experiment, allowing the simulation of the spatial and temporal variability of soil salinity during the irrigation experiment; ii) monitoring of $\sigma_a$ using a multi-coil CMD Mini-Explorer EMI sensor which takes $\sigma_a$ measurements over six different depth sensitivity ranges; iii) running numerical simulations by a physically-based hydrological model to study how well the EMI survey and time-lapse inversion can

resolve the $\sigma_b$ distribution in space and time in our experimental set up and to infer the best inversion parameters; iv) inverting time-lapse field $\sigma_a$ to map the spatio-temporal variability of $\sigma_b$ and to interpret them in terms of concentration distributions over time.

## 2. Material and Methods

### 2.1. Experimental set-up

The experiment was carried out at the Mediterranean Agronomic Institute of Bari in south-eastern Italy. The soil is classified as Colluvic Regosol, consisting of silty-loam material of an average thickness of 0.7 m lying on fractured calcarenite bedrock. Figure 1 shows the experimental set-up, consisting of four experimental plots each of 30 m length and 3.6 m width, equipped with a drip irrigation system with nine irrigation dripper lines placed 0.40 m apart and characterized by an inter-dripper distance of 0.20 m. Pressure self-compensating drippers were used and the Emission Uniformity was over 90%. The soil was bare during the experiment period to avoid the effect of root uptake on the interpretation of the results. The four experimental plots were irrigated with water at four different salinity levels. Experimental plot 1: water at 1 dSm$^{-1}$ (hereafter referred to as P1); experimental plot 2: water at 4 dSm$^{-1}$ (hereafter P2); experimental plot 3: water at 8 dSm$^{-1}$ (hereafter P3); experimental plot 4: water at 12 dSm$^{-1}$ (hereafter P4).

The soil-bedrock depth was measured in 40 points by augering and the resulting spatial distribution is shown in Fig. 2. An apparent variability in the depth to bedrock was revealed, which may have significant impacts on spatio-temporal distribution of water and solute concentration.

Irrigations were started on 30 September 2016 and all experimental plots were irrigated 12 times until 21 November 2016 with the same amount of water and delivering day. Water salinity was induced by adding calcium chloride (CaCl$_2$) to well water having a salinity of about 1 dSm$^{-1}$. Twelve injections of saline water were applied. Water salinity was induced by adding calcium chloride (CaCl$_2$) to well water having a salinity of about 1 dSm$^{-1}$. Each saline application was of about 18 mm with a Cl$^-$ concentration of about 0.1 mmol cm$^{-3}$. The volumes of supplied water were calculated according to the differences between two consecutive level measurements in a Class A evaporimeter. The details of irrigation events and precipitation information are given in Fig. 3. EMI measurements were taken four times during the experiment period in each experimental plot along three transects, 1.2 m apart and at 2 m spacing, as shown in Fig. 1. All measurements were taken 1-3 days after the irrigations, allowing relatively time-stable water contents to be assumed at each site throughout the monitoring phases.

### 2.2. EMI analysis

#### 2.2.1. Characteristics of the EMI sensor

$\sigma_a$ data were collected using a CMD Mini-Explorer device (GF Instruments, Brno, Czech Republic). The characteristics of this device make it especially suitable for monitoring electrical conductivity at relatively shallow depths. The CMD Mini-

Explorer was used to measure $\sigma_a$ in VCP (vertical coplanar, i.e. horizontal magnetic dipole configuration) mode and then HCP (horizontal coplanar, i.e. vertical magnetic dipole configurations) mode by rotating the probe 90° axially to change the orientation from VCP to HCP mode. The probe has three receiver coils with 0.32 ($\rho32$), 0.71 ($\rho71$) and 1.18 ($\rho118$) m distances from the transmitter coil and operates at 30 kHz frequency. With the largest coil spacing, the instrument has an effective depth investigation of 1.8 m in the HCP mode and 0.9 m in the VCP mode.

### 2.2.2. Electromagnetic Forward Model

The electromagnetic forward modelling is solved by applying the full solution of the Maxwell equations to calculate the $\sigma_a$ responses of a 1D model. The response of a layered media (secondary magnetic field) excited by a small horizontal transmitter loop, above the ground surface can be expressed in terms of Hankel transform (Zhang et al., 2000):

$$\mathbf{H_z} = \frac{\mathbf{M}}{\mathbf{4\pi}} \int_0^\infty \left[\mathbf{R_{TE}}\mathbf{e}^{-\beta(2z_s - z_r)}\right]\boldsymbol{\beta}^2\mathbf{J_o}(\boldsymbol{\beta}\mathbf{r})\mathbf{d}\boldsymbol{\beta}, \qquad (1)$$

where M is the moment of the transmitter loop emitting at an angular frequency $\omega$, $z_s$ and $z_r$ are the heights of the transmitter and receiver loops, respectively. $r$ is the transmitter-receiver distance and $J_o$ is the Bessel function of first kind and order zero.

The kernel function $R_{TE}$ is defined as:

$$\mathbf{R_{TE}} = \frac{\mathbf{Z}^1 - \mathbf{Z_0}}{\mathbf{Z}^1 + \mathbf{Z_0}}, \qquad (2)$$

where $Z_0$ is the intrinsic impedance of free space, $Z^1$ is the input impedance at the first layer calculated by a recursive procedure. Similar equations can be written for vertical coplanar loops. In such case the secondary magnetic field is:

$$\mathbf{H_y} = \frac{\mathbf{M}}{\mathbf{4\,\pi\,r}} \int_0^\infty \left[\mathbf{R_{TE}}\mathbf{e}^{-\beta(2z_s - z_r)}\right]\boldsymbol{\beta}\mathbf{J_1}(\boldsymbol{\beta}\mathbf{r})\mathbf{d}\boldsymbol{\beta}, \qquad (3)$$

where $J_1$ is the first-order Bessel function.

$\sigma_a$ (mSm$^{-1}$) is usually calculated assuming the low induction number (LIN) approximation using the formula:

$$\boldsymbol{\sigma_a} = \frac{\mathbf{4000}}{\boldsymbol{\omega}\boldsymbol{\mu_o}\,\mathbf{r}^2}\left(\frac{\mathbf{H_s}}{\mathbf{H_p}}\right)_{\mathbf{Q}}, \qquad (4)$$

where $\mu_0$ is permeability of free space and Q denotes the out-phase component of the secondary to primary magnetic field coupling ratio.

### 2.2.3. Time-lapse inversion

With the time-lapse inversion, one seeks to calculate the temporal variation of the conductivity along a transect. The quasi-
2D inversion algorithms based on Monteiro Santos (2004) with a modification of the algorithm proposed by Kim et al. (2009) were used in this study. The problem is resolved in both algorithms iteratively starting from a uniform model. Two different levels of constraints, S1 and S2, were applied. In the S1 option, the corrections to the model parameters at each iteration are calculated by solving the system of equations:

$$(J^T J + \lambda C^T C + \alpha M^T M)\delta\vec{p} = J^T\vec{b} - \alpha M^T M \vec{p}, \tag{5}$$

In the S2 option, the corrections of the parameters at each iteration are calculated solving the equations:

$$(J^T J + \lambda C^T C + \alpha M^T M)\delta\vec{p} = J^T\vec{b} + \lambda C^T C (\vec{p} - \vec{p_o}) \, \alpha M^T M (\vec{p} - \vec{p_o}), \tag{6}$$

where $\delta p$ is the vector comprising corrections of the parameters (logarithm of conductivities, $p_j$) of an initial model; $p_o$ refers to a reference model; b is the vector containing the differences between the logarithm of the observed and calculated apparent conductivities. J is the Jacobian matrix with elements given by $\frac{\sigma_j}{\sigma_{ai}^c}\frac{\partial\sigma_{ai}^c}{\partial\sigma_j}$. $\lambda$ is a Lagrange multiplier and determines the amplitude of the parameter corrections in the space domain and the regularisation matrix C stores the coefficients of the spatial roughness of the model parameters at time t which is defined as:

$$\delta r_j = \delta_{PjE} + \delta_{PjW} - 4\delta_P + \delta_{PjN} + \delta_{PjS}, \tag{7}$$

where the elements of matrix C are 1 or -4 according to the position of the neighbours. $\alpha$ is a parameter that determines the amplitude of the parameter corrections in the time domain, and M is a square matrix that accounts for the temporal continuity of the model parameters.

The elements of matrix M are defined in terms of models at time t-dt and t+dt. The model Misfit is calculated using the following equation:

$$\text{Misfit} = \sqrt{\frac{1}{N}\sum_{i=1}^{N}\ln(\sigma_a^o - \sigma_a^c)^2}, \tag{8}$$

where N is the number of apparent conductivity value, with $\sigma_a^o$ and $\sigma_a^c$ a representing observed and calculated $\sigma_a$, respectively.

In this algorithm, two regularizations are imposed in both space and time domains. Consequently, the spatial and temporal Lagrangian multipliers have to be optimized. The spatial Lagrangian multiplier ($\lambda$) controls the relative importance of spatial model smooth and data-response misfit and decreases gradually during the inversion process to resolve more detailed model parameters. Larger $\lambda$ tends to generally produce a model with a larger misfit error but smooth variation of conductivity values. Larger $\lambda$ is usually acceptable if the soil conductivity changes in a smooth manner and allows producing a model that is reasonably more realistic. In contrast, a smaller $\lambda$ is usually required when sharper soil conductivity changes are expected in order to resolve the sharp boundaries. A suitable $\lambda$ value is usually determined empirically based on the expected distribution of $\sigma_b$ and by performing inversions with different values. The second regularization, temporal Lagrangian multiplier ($\alpha$), is the temporal damping factor that gives the weight for minimizing the temporal changes in the conductivity along the time axis. $\alpha$ is a constant value and is defined by the similarity of the two consecutive reference times. The larger the $\alpha$ value, the more similar are the reference models that result from the inversion; a value of zero means no temporal constraints are applied (i.e. a traditional non-time-lapse inversion). The misfit function in this algorithm is the square root of the sum of the squares of the differences divided by the number of the measurements and is expressed in $mSm^{-1}$.

### 2.3. Synthetic experiment

#### 2.3.1. Hydraulic properties data set

A large dataset of hydraulic properties was already available from the experimental site, which was obtained by previous laboratory and field hydraulic characterizations carried out during several measurement campaigns (e.g. Coppola et al., 2011a, b; 2013; 2015). Seventy soil samples were collected from the Ap and Bw horizon in the experimental farm. Saturated hydraulic conductivity, $K_0$, and water retention experimental data were measured in the laboratory by the falling head permeameter (Reynolds and Elrick, 2003) and tension table method (Dane and Hopmans, 2003), respectively. The water retention data were fitted to the van Genuchten model (van Genuchten, 1980):

$$S_e = \frac{\theta - \theta_r}{\theta_0 - \theta_r} = [1 + |\alpha h|^n]^{-m}, \qquad\qquad h < 0 \qquad\qquad (9)$$

$$\theta = \theta_0, \qquad\qquad h = 0 \qquad\qquad (10)$$

where $S_e$ (-) is the effective saturation, $\alpha$ (cm$^{-1}$), $n$ (-) and $m$ (-) are shape parameters, $\theta_0$ (-) and $\theta_r$ (-) are the saturated and residual water content, respectively.

The hydraulic conductivity was estimated by the van Genuchten-Mualem model (van Genuchten, 1980), with $m=1-1/n$:

$$K_r(S_e) = \frac{K(S_e)}{K_0} = S_e^\tau \left[1 - \left(1 - S_e^{\frac{1}{m}}\right)^m\right]^2, \qquad\qquad (11)$$

where $K_0$ is the saturated hydraulic conductivity, $K_r$ (-) is the relative hydraulic conductivity, and $\tau$ (-) is a parameter which accounts for the dependence of the tortuosity and the correlation factors on the water content.

Due to the different hydrological behaviour between laboratory and field (Kutilek and Nielsen, 1994), the laboratory-derived curves were scaled to the field curves by applying the procedure described in Basile et al. (2003; 2006). Statistics of the parameters are reported in Table 1. The data in the table indicate a larger variability of the hydraulic properties of the Bw horizon, compared to the Ap horizon. This is probably due to the frequent roto-tillage of the Ap horizon, inducing significantly greater homogeneity of this soil layer. Actually, even the apparent larger variability of the $K_0$ for the Ap horizon is quite lower than the variability reported in the literature, which is generally characterized by coefficients of variation (CV) much larger than 100% (Kutilek and Nielsen, 1994; Mallants et al., 1996; Coppola et al., 2011) and may even reach values of 450% (Carsel and Parrish, 1988). The hydraulic properties parameters for the bedrock were those determined on the same type of bedrock by Caputo et al. (2010; 2015).

#### 2.3.2. Synthetic hydrological simulations

2D numerical simulations of the infiltration and redistribution process were carried out using Hydrus 2D/3D software (Šimůnek et al., 2016), by introducing the actual boundary conditions imposed during the experiment to: i) gain an insight to

the spatio-temporal distribution of water and solute concentration during the experiment period, prior to interpreting the field
EMI data and to ii) optimize the EMI inversion parameters. The Richards Equation and Advection-Dispersion Equation were
used to simulate water flow and solute transport, respectively. The distributions of the hydraulic parameters (see Table 1)
were used to generate several synthetic transects, with variable hydraulic properties and depth to bedrock (electrically
resistive layer), in order to simulate different distributions of water contents, solute concentrations and soil depths and to
explore their role on the variability of the $\sigma_a$ response. Each synthetic transects consisted of an assembly of 30 interacting
soil columns, each with its own hydraulic properties and depth to bedrock. Each of the 30 soil profiles included three
horizons: Ap (0-15 cm), Bw and bedrock. The Bw and bedrock thickness changed in each soil profile according to the depth
of the soil-bedrock interface. For hydraulic properties, the transect variability was assumed to be characterized by the
variability of the parameters $\theta_0$, $\alpha$, n and $K_0$. The hydraulic properties and depth to bedrock were assigned randomly to each
soil column by using the statistical distribution of each parameter (see Table 1). Statistical tests showed that $\theta_0$ and n were
normally distributed, $K_0$ and $\alpha$ log-normally distributed, and depth to bedrock uniformly distributed. The means and the
covariance matrix for $\theta_0$, log $K_0$, log $\alpha$, and n were computed. Due to the continuous ploughing and other tillage practices
(see Sect. 2.1), the Ap horizon is rather homogeneous and therefore only the hydraulic parameters of the horizon Bw were
considered to vary stochastically. The hydraulic parameters of both Ap horizon and bedrock were fixed to their average
values.

20 synthetic transects with 30 random vectors of the four parameters ($\theta_s$, $\alpha$, n and $K_0$) were produced from the correlated
multivariate distribution by generating a vector x of independent standard normal deviates and then applying a linear
transformation of the form $x=m+Lr_n$, where m is the desired vector of means and L is the lower triangular matrix derived
from the symmetric covariance matrix $V=LL^T$ decomposed by Cholesky factorization (Carsel and Parrish, 1988). $\theta_r$ was set
to zero for all simulations. For solute transport, a longitudinal dispersivity of 2 cm was assumed according to a previous
experiment carried out in the same field (Coppola et al., 2011b). Transverse dispersivity was assumed to be one tenth of the
longitudinal dispersivity (Mallants et al., 2011). The simulation time – according to the field experiment data – included a
period of 91 days from 01 September 2016 to 30 November 2016 in which 12 irrigations were applied (a total of 210 mm),
having the water an EC of 12 dSm$^{-1}$ (each one of about 18 mm with a Cl$^-$ concentration of about 0.1 mmol cm$^{-3}$). The 12
dSm$^{-1}$ was considered because spatial and temporal variabilities of $\sigma_b$ were expected to be larger and more apparent due to
greater salinity changes. Potential evapotranspiration was estimated by a local agrometeorological station; irrigation fluxes
and Cl$^-$ concentration in the irrigation water were considered as top boundary conditions. Free drainage was assumed at the
lower boundary (z=-150 cm). Initial soil water content value was set to 0.25 cm$^3$ cm$^{-3}$, based on the field measurements by
TDR probe, and the initial Cl$^-$ concentration was set equal to zero.

It is worth noting that the use of synthetic transects does not aim to address the overall spatial variability of soil properties
potentially observable in the investigated field, but to randomly select a reasonable number of different scenarios to better
understand how solute concentration and water content changes during the experiment influence $\sigma_b$ distribution. They also
help to identify a proper regularization strategy to invert measured $\sigma_a$ data in the field. In this sense, the number (20) of

transects is a trade-off between the need of accounting for the possible heterogeneity of the field and the computational challenge of carrying out too many synthetic data inversions.

For each generated transect, numerical simulations produced distributions of water contents and Cl- concentrations.

## 2.4.  Site-specific calibration $\theta$-$\sigma_w$-$\sigma_b$

The water content and Cl- concentration distributions were converted to bulk electrical conductivity ($\sigma_b$) distributions by using the model proposed by Malicki and Walczak, 1999:

$$\sigma_w = \frac{\sigma_b - a}{(\varepsilon_b - b)(0.0057 + 0.000071S)} ,  \tag{12}$$

where $\varepsilon_b$ (-) is the dielectric constant, which is related to the water content, and $\sigma_w$ (dSm$^{-1}$) is the electrical conductivity of the soil solution. The latter was obtained by using a linear relationship C-$\sigma_w$ for solutions at different concentrations of calcium chloride.

The parameters for the Malicki and Walczak model were calibrated through a laboratory experiment. Specifically, $\varepsilon_b$ and $\sigma_b$ for different values of $\sigma_w$ were simultaneously measured on reconstructed soil samples values using TDR probes. For this

purpose, four PVC cylinders (8 cm in diameter and 15 cm height) were filled with air-dried soil reaching a dry bulk density of about 1.1 g cm$^{-3}$, imitating the field condition. Each soil sample was wetted by adding 10 ml of CaCl$_2$ solution at a specified electrical conductivity: 1, 2, 4 and 6 dSm$^{-1}$, respectively. The cylinders were covered by 0.05-mm plastic foil before the measurement in order to avoid evaporation and to equilibrate with the air temperature of 20°C. The procedure was repeated 16 times for each soil sample to measure soil water content values ranging from air-dry to near saturation. For each

wetting step, the measurements of $\theta$ and $\sigma_b$ were carried out using TDR three-wire probes (10 cm long with a rod diameter of 0.3 cm and rods spaced 1.2 cm) vertically inserted in the soil columns. The following parameters of Eq. (12) were obtained: $a = 3.6$ dSm$^{-1}$; $b = 0.11$.

## 2.5.  Schematic view of the approach used in the paper

The logical sequence of the different steps used in the proposed approach is described in Fig. 4

1.  Starting from the statistical distribution of the hydraulic properties and the physical characteristics of the system under study (the latter limited to only depth to the bedrock, in this specific case), as described in Sect. 2.3.1., several synthetic transects are generated, each accounting for some of the variability in terms of hydraulic properties and physical characteristics that the EMI sensor potentially experience during the monitoring;

2.  Hydrological simulations for each of these synthetic transects are thus carried out, each producing synthetic distributions
of both water contents and solute concentrations, as described in sec 2.3.2 which, in turn, is converted to as many synthetic distributions of $\sigma_b$ by using a specifically developed $\theta$-$\sigma_b$-$\sigma_w$ calibration relationship, discussed in Sect 2.4;

3.  These $\sigma_b$ distributions are used in a forward EMI modelling procedure (see Sect. 2.2.2.) to generate synthetic $\sigma_a$ data for all the synthetic transects considered;

4. The $\sigma_a$, $\theta$ and $\sigma_w$ (C) distributions is used to guide the inversion of EMI readings obtained during real monitoring campaigns carried out in the physical system under study in two different (but related) ways: i) to have an a-priori knowledge of where a measured EMI reading may come from (e.g. at a given time, a measured $\sigma_a$ distribution could come from the depth of the water or solute propagation front, from the depth to the bedrock, from water accumulation at the soil-bedrock interface); ii) to identify the optimal regularization parameters, discussed in sec 2.2.3., to be used in the inversion model of the real EMI data, by looking for example to the parameter value allowing for a satisfactory results for most of the synthetic transects. For the latter purpose, we compare different inversion parameters combinations in terms of correlation (correlation coefficient, R), precision (mean square error, RMSE) and bias (mean error, ME) between the simulated and modelled $\sigma_b$;

5. Finally, with all this information at hand, one is ready to look at the real EMI datasets, by producing more realistic $\sigma_b$ distributions which can now be seen with much more confidence compared to cases where EMI inversion has to be carried out without any prior information from the experiment and, thus, producing less interpretable and more uncertain $\sigma_b$ distributions.

The Results and Discussion section below will show the application of the approach to the system under study, with an analysis of the real EMI data carried out only in the final phase, when having available all the information needed to guide the inversion of the real EMI data and interpretation of the obtained models.

## 3. Results and Discussion

### 3.1. Synthetic spatio-temporal distributions of solute water content and solute concentration

This section is to show how the synthetic simulations described in the section 2.3.2 can be used to analyse the sensitivity of the EMI response to both the hydrological behaviour and physical characteristics of the system under study.

As an example, Fig. 5 depicts the depth to the bedrock for one of the synthetic transects described in Sect. 2.3.2. For the same transect, Fig. 6 shows, for four selected days, the spatial distribution of water content obtained from the Hydrus 2D/3D simulations with irrigation water at 12 dSm$^{-1}$. The soil shows very high values of water content on the selected days. This is because the water was supplied to the soil only 1-3 days before the selected days. On the other hand, lower values were obtained at larger depths, within the bedrock. The water content distributions show a significant lateral heterogeneity while the temporal variations are very small. The spatial variations of water content distribution can be partly explained by the variability of the hydraulic properties. However, soil depth proved to be the dominant factor in determining water content lateral variability. In fact, high inverse correspondence of soil depth with water content is revealed in the vertical profiles where bedrock is superficial (e.g. profiles 3, 4 and 5) or deep (i.e., profiles 19, 20 and 21). The correlation between soil depth and water content averaged along each profile is very high (r=0.88) for the four dates, thus confirming that the depth of the bedrock is the main factor governing the water distribution.

In terms of the temporal variations, the water content distribution did not change significantly and the average soil water content of the whole soil profile was found to be similar in the selected dates, with an average in the range 0.20-0.22 cm$^3$ cm$^{-3}$ and a standard deviation of 0.02 cm$^3$ cm$^{-3}$. This was expected as the experimental field was irrigated regularly and the four selected dates refer to approximately the same time after an irrigation application (1-3 days). However, small differences of water content may be observed near the soil surface, which may well be explained by the evaporation process taking place during the 1-3 days after irrigation and mostly involving the shallower soil layer. For example, the near-surface shows higher water content (i.e. 0.25-0.30 cm$^3$ cm$^{-3}$) on 26 Oct, because the irrigation took place one day before on 25 Oct. On the other hand, lower water content is evident in the near-surface on 17 Oct due to a three days gap between the water irrigation on 14 Oct and simulation date. Figure 7 shows the spatio-temporal distribution of Cl$^-$ concentration obtained for the same synthetic transect of Fig. 6. Compared to the water content, Cl$^-$ concentrations show a significantly greater evolution over time, with a slow and steady Cl$^-$ concentration increase along the soil profile due to the twelve injections of saline water. On average, the front of chloride deepens slowly at a fairly constant rate, but with a lateral variability which is related to the lateral variability of water contents and hydraulic properties. The lateral variability is quite low in the first days of solute applications and becomes evident only when the solute migrates deeper into the soil profile. The lateral variability of Cl$^-$ concentration is mainly related to the depth to bedrock, as can be immediately observed by comparing the solute distributions and the transect morphology shown in Fig. 5. The depth of the soil-bedrock interface as well as soil hydraulic properties conditions the spatial distributions of water contents, which, in turn, influence the Cl$^-$ concentration distribution.

### 3.2. Simulated spatio-temporal distribution of $\sigma_b$

Figure 8 shows the spatio-temporal distribution of $\sigma_b$ for the same synthetic transect, obtained by converting water contents and solute concentrations by applying Eq. (12). The Fig. 8 shows a resistive zone beneath a conductive zone. The conductivity of the resistive zone varies slightly spatially and temporally, however the conductivity of this zone is generally within 25 mSm$^{-1}$. The resistive zones in the maps correspond to the bedrock in the study area (see Fig. 5). The conductivity of the upper layers changes significantly both spatially and temporally from an average conductivity of 50 mSm$^{-1}$ on 17 Oct to more than 100 mSm$^{-1}$ on 23 Nov. The time evolution of the conductive zone is evident and, as the water content does not change significantly over time, is mostly related to the chloride propagation during the simulation. The lateral variation at any time, by contrast, is largely ascribable to bedrock topography.

### 3.3. Time-lapse synthetic $\sigma_a$ data

The spatio-temporal distribution of $\sigma_b$ shown in Fig. 8 was used to generate the time-lapse synthetic $\sigma_a$ data. The generated $\sigma_a$ data for the model obtained on 23 Nov (see plot d in Fig. 8) is shown in Fig. 9. Profile $\rho32$ shows the greatest $\sigma_a$ values in each orientation, while the minimum $\sigma_a$ was recorded on profile $\rho118$ indicating a conductive zone over a resistive zone, which is expected from the model shown in Fig. 8d. In addition, significant lateral $\sigma_a$ change is evident along the transect with strongest fluctuations in $\rho32$ in both orientations. This is because of the strong lateral $\sigma_b$ variations at near surface due

to the saline water infiltration and heterogeneity of the subsurface. The general behaviour of the synthetic $\sigma_a$ data suggests that the field data might have strong lateral $\sigma_a$ changes along the transect and a careful processing of data is required (e.g. filtering of $\sigma_a$ data should be avoided).

### 3.4. Optimizations of the inversion parameters

In this section we show how well the developed inversion method can resolve the $\sigma_b$ distribution from generated synthetic $\sigma_a$ distribution. Firstly, we investigated the influence of the spatial smoothing parameter ($\lambda$) and the inversion algorithm S1 (Eq. (5)) and S2 (Eq. (6)) in resolving the $\sigma_b$ distribution. In this regard, the generated $\sigma_a$ data were inverted using both S1 and S2 and different values of $\lambda$ in the 0.01 to 10 range. Figure 10 shows an example of the $\sigma_b$ distribution models after inverting the synthetic $\sigma_a$ data presented in Fig. 9 This example was selected due to the larger lateral and vertical contrast.

Ideally, the obtained $\sigma_b$ distribution should be very similar to the one shown in Fig. 8d. However, looking closely at the obtained $\sigma_b$ distribution, we observe that all of them are different, to some extent, to the one shown in Fig. 8d. First of all, we note that the model obtained using $\lambda$ values greater than 1 (not shown here) and regardless of the choice of inversion algorithm (i.e. S1 and S2) shows a high misfit error and also significantly over-smoothed $\sigma_b$ distribution. This is not surprising as the sharp vertical spatial variability of $\sigma_b$ due to the saline water irrigation, soil heterogeneity as well as the shallow bedrock cannot be well resolved using high $\lambda$ (over-smoothed parameters). Consequently, a high $\lambda$ is not a wise choice when sharp vertical and lateral conductivity contrasts are expected in the field. The obtained model using S2 algorithm and moderate $\lambda$ values (i.e. 0.5) also yields a very smooth model. In contrast, the obtained models using S1 - $\lambda$=0.05, S1 - $\lambda$=0.5 and S2 - $\lambda$=0.05 do a better job in resolving near-surface anomalies. However, the conductivity distribution at depth is not well recovered using the S1 algorithm.

In Fig. 11, we plotted the synthetic $\sigma_b$ distributions data against the obtained modelled $\sigma_b$ distributions using different inversion parameters shown in Fig. 10 and calculated the statistical scores R, RMSE and ME to further investigate the impact of the inversion algorithm and parameters in resolving the $\sigma_b$ distributions. Firstly, we observed that both S1 and S2 algorithms underestimated the $\sigma_b$ to some extent, judging from the ME values. The S2-$\lambda$=0.5 and S1-$\lambda$=0.05 show weaker statistical scores with higher RMSE and ME and relatively lower R. The over-smooth impact of S2-$\lambda$=0.5 and the resulting highest ME indicates that the S2 algorithm with moderate to high $\lambda$ values is not a rational choice when large spatial and vertical variations of $\sigma_b$ are expected. On the other hand, the lowest R and highest RMSE, obtained for S1-$\lambda$=0.05, suggest that the S1 algorithm with very small values of $\lambda$ is not able to well predict the spatial variation of $\sigma_b$. The use of the S1 algorithm with very small values of $\lambda$ usually apply insignificant spatial constraint that may result in more inconsistency between synthetic $\sigma_b$ and modelled $\sigma_b$ distributions. The S2-$\lambda$=0.05 and S1-$\lambda$=0.5 show better statistical results with higher R and lower RMSE and ME which make them a better choice for the inversion of field data. While both inversion parameters set present almost the same RMSE, the S2-$\lambda$=0.05 yields a higher R, suggesting that the S2-$\lambda$=0.05 can better resolve the spatial $\sigma_b$ distributions. This is expected from a comparison of S2-$\lambda$=0.05 and S1-$\lambda$=0.5 results in Fig. 10 and Fig. 11 where a relatively lower correlation is evident between synthetic $\sigma_b$ and modelled $\sigma_b$ at lower ranges of $\sigma_b$ (Fig. 11) located at depth

more than 70 cm (Fig. 10 and Fig. 11) when we used the S1-$\lambda$=0.5. The difficulty of resolving a resistive zone at depth and beneath a conductive zone is indeed expected. In fact, the sensitivity of the EMI signals is very limited over the resistive zone and therefore the resistive zone cannot be well resolved. The condition will be worse in our study as a resistive zone is located beneath a conductive zone: the EMI response of the subsurface will be dominated by the influence of the near surface conductive zone. In addition, five of the six depths of investigation of the CMD Mini-Explorer are limited to the first

1 m and, as a result, a lower resolution is expected at greater depths. The S2 algorithm did a better job in resolving the resistive zone at depth. This is because the S2 algorithm constrains the value of each cell to the reference model during the inversion process which limits the large variations of each cell during the inversion process.

In terms of recovering the absolute values of soil electrical conductivity, it appears that all obtained models underestimate the conductivity of the anomalies near the surface. This can be explained by over-parameterized inverse problem and the

effects of smoothing from regularization applied in the inversion algorithm, as well as the impact of the resistive zone beneath the conductive zone. In addition, the 6 measurements per site with site spacing of 1 m is not sufficient for recovering sharp $\sigma_b$ variability along the transect. Measuring $\sigma_a$ at different heights as well as smaller site spacing enables more $\sigma_a$ data to better resolve changes which occur over short depth and length increments.

In terms of the influence of the temporal smoothing parameter ($\alpha$), we explored different values (results not shown here). We

noticed that the values larger than 0.1 over-smoothed the expected temporal variation and thus the detailed variations cannot be resolved. We conclude that, for our study, a value of 0.05 is the best choice for $\alpha$ in resolving the temporal variation of $\sigma_b$. We repeated the same analyses using 20 different synthetics transects, consisting of a random assembly of 30 interacting soil columns, each with its own hydraulic properties and depth to bedrock and different saline treatment (results not shown here). The results of our analysis show that the algorithm S2 is the best choice in the presence of a resistive zone at depth and small

values of $\lambda$ and $\alpha$ work best for dealing with the sharp lateral and temporal expected changes that occurred during the experiment. Based on the synthetic tests, the spatio-temporal algorithm S2 described in Eq. (6) with $\lambda$ and $\alpha$ values of 0.05 and 0.05 respectively were selected to invert time lapse actual $\sigma_a$ datasets measuring during the experiment period.

### 3.5. Inversion of the real time-lapse $\sigma_a$ field data

Using the optimized inversion parameters obtained in Sect. 3.4., we inverted time-lapse $\sigma_a$ data collected over the four

experimental plots, P1, P2, P3, and P4. Fig. 12 shows the $\sigma_a$ data of the middle transect in each experimental plot (Fig. 1) referred to the last date of monitoring, i.e. 23 Nov. The $\sigma_a$ data shows a relatively similar pattern in both VCP and HCP modes with greater $\sigma_a$ values on $\rho$32 and $\rho$71 and the minimum $\sigma_a$ values recorded on $\rho$118, indicating a conductive zone over a resistive zone. In addition, greater lateral $\sigma_a$ changes in the VCP mode are evident along the four transects with noticeable fluctuations in P4, suggesting greater lateral $\sigma_b$ variations at near surface. The $\sigma_a$ data obtained from plot P1 and

P2 show lower range of conductivity, varying in the 20-40 mSm$^{-1}$ and 20-50 mSm$^{-1}$ ranges, respectively. P3 and P4 represent higher range of conductivity with the former in the 20-70 mSm$^{-1}$ range and the latter in the 20-80 mSm$^{-1}$ range.

Figure 13 shows the time-lapse inversion results for the same transects for four dates: 17 and 26 Oct; and 14 and 23 Nov 2016. The same colour scale was used for all models, allowing comparison of spatio-temporal variation of $\sigma_b$ along all profiles at different times. The corresponding model responses for the last measurements on 23 Nov are shown in Fig. 12.

The misfit errors for the P1, P2, P3, and P4 are 1.43, 1.21, 2.15 and 2.95 mSm$^{-1}$, respectively, indicating a good fit between data and model responses. Slightly higher misfit errors for P3 and P4 are probably due to greater range of $\sigma_a$ as well as the larger lateral variations of $\sigma_a$. Looking at EMI models at different times and along all four experimental plots, we identify two distinct zones: a resistive zone at depth more than 50 cm beneath a conductive zone. The conductivity of the resistive zone varies laterally and temporally along all four transects; however, the conductivity of this zone rarely exceeds 25 mSm$^{-1}$.

The resistive zones in the maps shown correspond to the bedrock in the study area (Fig. 2). The conductivity of this zone is slightly lower in P3 and P4 which is probably due to the shallower bedrock in these plots. In contrast, the conductive zone near-surface shows significant spatio-temporal conductivity changes depending on the conductivity of the injected water.

The time-lapse models obtained from plot P1 show the minimum spatio-temporal conductivity changes in this zone with conductivity varying in the 30-60 mSm$^{-1}$ range. The P2 plot shows larger spatial and temporal conductivity changes

compared to P1, as expected, with conductivity varying between 30 and 80 mSm$^{-1}$. The conductivity of this zone decreases slightly on 14 Nov. This is probably due to the impact of heavy rain (31 mm) on 7 Nov, inducing salt leaching and reducing the soil conductivity. The P3 and P4 plots show stronger conductivity changes both spatially and temporally with conductivity varying between 50 and 100 mSm$^{-1}$. The first models, obtained for 17 Oct, shows the minimum conductivity among the four selected dates. This is consistent with the simulated distributions shown in Fig. 8 and it is probably due to the

saline front being still undispersed in the initial propagation phase (Fig. 7).

As the saline water is continuously added to the soil surface, the Cl$^-$ concentration increases in the soil and also deepens slowly in the soil profile, thus increasing soil electrical conductivity. Consequently, the higher soil conductivities seen on 23 Oct (Fig. 13, P3 and P4) are due to higher Cl$^-$ concentration and its propagation to deeper layers. The conductivity models obtained for 14 Nov show a decrease in soil conductivity in both P3 and P4, although there were 6 irrigations after 23 Oct.

On the other hand, the conductive zone extends to deeper soil. This is not surprising as we already expected it from the simulations (see Fig. 8). This behaviour is probably due to the rainfall event on 7 Nov (as noted on P2) which diluted the Cl$^-$ concentration in the soil. This suggests that the EMI surveys and data modelling detected well the expected change in Cl$^-$ concentration.

Finally, the models obtained for the data collected on 23 Nov, depicted in Fig. 13, show the maximum soil conductivity and

extension of the conductive zone among the selected dates in P3 and P4. A comparison of these models with our simulations results discussed in 4.1 and 4.2 show that the increase of Cl$^-$ concentration in soil after twelve irrigation events, as well as the redistribution of Cl$^-$ during the experiment period, are the main reason for increase in soil conductivity. Judging from the water content distribution maps, obtained from numerical simulations and shown in Fig. 6, we expect very small temporal variations of water content in all experimental plots. Consequently, the temporal variations of Cl$^-$ concentration and

distribution is expected to be the key factor in temporal variations of soil electrical conductivity.

From a management perspective, the discussion above suggests that using the described approach regularly after irrigation applications may allow monitoring of salt propagation and redistribution. As water content values and patterns are expected to be quite similar at similar times after each irrigation event, changes in bulk electrical conductivities obtained by an EMI sensor will be closely related to changes in salt concentrations.

**4. Conclusion**

In this study, we carried out a time-lapse EMI survey over four experimental plots irrigated with water at four different salinity levels during three months. We examined how well the time-lapse EMI measurements and a time-lapse inversion algorithm can be used to monitor soil salinity variability in space and time through performing simulation experiments and inversion processes. Based on our detailed simulations and synthetic tests as well as the interpretation of the time-lapse models, the following main conclusions can be drawn:

1. The numerical simulations performed in this study allow predictions of the spatial and temporal variability of soil salinity and water content during the irrigation experiment prior to the modelling of field $\sigma_a$ data. This improves our understanding of how soil salinity and water content changes during the experiment influence $\sigma_b$ distribution in time, which may be crucial for interpreting EMI models in terms of soil salinity and water content distribution. They also provided a synthetic time-lapse EMI dataset very similar to the field condition, allowing the optimization of data processing and to find the best inversion approach for the field experiment. From the synthetic analysis, we also established that the EMI measurements do not return enough subsurface information to resolve the expected sharp conductivity changes in this specific experiment.

2. A comprehensive investigation of our results and joint-interpretation of numerical simulations and time-lapse models reveals that the soil Cl$^-$ concentration change is the key factor responsible for $\sigma_b$ changes when the EMI surveys are repeated after the irrigation at the same time. In fact, repetition of the EMI surveys after the irrigation has three main advantages: i) the soil is wet and conductive and, in such a condition, the signal/noise ratio is usually better and EMI data will be more reliable; ii) the water content distribution will not change significantly, allowing to better study the impact of soil salinity changes in time; iii) The sensitivity of $\sigma_b$ to the Cl$^-$ concentration in wet soils is higher and thus EMI inversion results may be used to interpret salt propagation with more confidence.

3. A previously developed least-squares 4-D space–time domain inversion algorithm was implemented in this study to invert entire time-lapse EMI data sets simultaneously for monitoring soil salinity for the first time. The regularizations in this algorithm are introduced in both space and time domains to improve the stability of the inversion process and to reduce the inversion artefacts. Using a synthetic test, the applicability of the algorithm has been examined and we showed that this approach can provide information about the conductivity variability in space and time. More synthetic tests are required to further investigate the efficiency of the inversion algorithm under different scenarios, however, we anticipate that the algorithm can be used for other EMI monitoring surveys.

4. Performing EMI surveys over four experimental plots irrigated with water at four different salinity levels, we evaluated the potential of EMI surveys for monitoring the dynamics of soil salinity due to irrigation. Comparing the EMI models obtained from four experimental sites, we showed that the $\sigma_b$ variations are consistent with the expectations related to the amount of salt and water irrigated at each plot during the experiment. The water content did not change significantly during the EMI measurement campaigns, hence, the temporal variations of $\sigma_b$ as well as their difference at each plot are mainly related to the soil salinity distributions. In other words, the results indicate that the applied experimental methodology is strongly capable of giving information on spatial and temporal variability of soil salinity. Further investigations have to be conducted to use EMI sensors for monitoring salinity under significant water content changes over time. In this case, discriminating the role of water content changes and salinity changes on the $\sigma_a$ response may be quite complicated if not impossible.

5. The EMI method provides enormous advantages over traditional methods of soil sampling because it allows in-depth and non-invasive analysis, covering large areas in less time and at a lower cost. However, a proper interpretation of the EMI inversion models in terms of soil process is usually difficult owing to the fact that the soil electrical conductivity is a complex function of soil properties, which may vary significantly over space and time. Thus, retrieving soil properties from EMI data requires appropriate understanding of site-specific soil processes. Our study shows that the independent interpretation of time-lapse EMI data without hydrologic insight and understanding of soil processes may be misleading. This fact highlights the necessity of collaboration of geophysicists, soil scientists and hydrologists to construct a hydrologic conceptual model which can explain the salinity and water process.

**Author contribution**

Conceptualization: MF, AB, AC. Data curation: RDM, GD. Formal analysis: MF, DA. Funding acquisition AB, FMS. Investigation: RDM, GD. Project administration: AB, GD, AC. Software: FMS, MF, DA. Writing – original draft preparation: MF. Writing – review & editing: MF, DA, AB, ABin, AC.

**Competing interests**

The authors declare that they have no conflict of interest.

**Acknowledgements**

This research was performed within projects SALTFREE: "Salinization in irrigated areas: risk evaluation and prevention" (supported by the Italian Ministry of Agricultural, Food and Forestry Policies, grant no. D.M. 28675/7303/15 and by the Portuguese research agency, Fundação para a Ciência e a Tecnologia (FCT), grant no. ARIMNET2/0004/2015), and

SOIL4EVER: "Sustainable use of soil and water for improving crops productivity in irrigated areas" (supported by FCT, grant no. PTDC/ASP-SOL/28796/2017). This publication is supported by FCT – project UIDB/GEO/50019/2019/IDL. Mohammad Farzamian was supported by a contract within project SOIL4EVER.

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

**Table 1: Statistics (mean and coefficient of variation - CV) of the hydraulic property parameters of the two soil layers and bedrock.**

| Layer | | $\theta_r$ (cm$^3$ cm$^{-3}$) | $\theta_s$ (cm$^3$ cm$^{-3}$) | $\alpha$ (cm$^{-1}$) | n (-) | $K_0$ (cm d$^{-1}$) | $\tau$ (-) |
|---|---|---|---|---|---|---|---|
| Ap | Mean | 0.000 | 0.329 | 0.070 | 1.40 | 10.30 | 0.5 |
| | CV% | | 4.6 | 15.7 | 7.9 | 64.2 | |
| Bw | Mean | 0.000 | 0.315 | 0.025 | 1.38 | 33.42 | 0.5 |
| | CV% | | 13.2 | 78.2 | 11.5 | 240.2 | |
| Bedrock | | 0.068 | 0.354 | 0.055 | 3.67 | 19.02 | 0.5 |


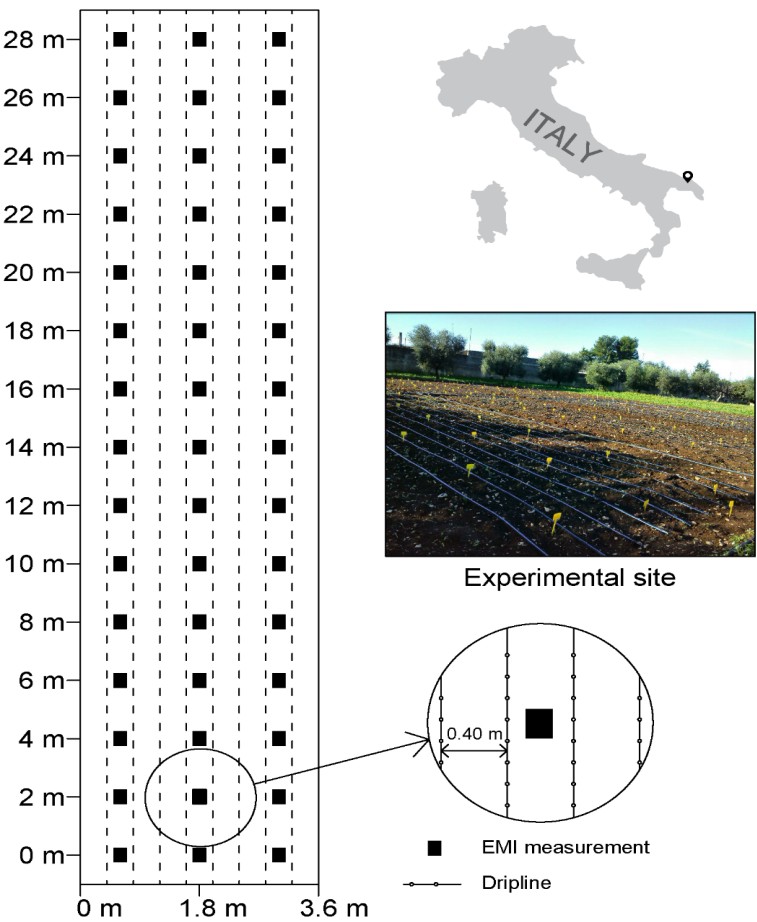

**Figure 1: A schematic display of the experimental setup. Four experimental plots were designed equally and irrigated with the same amount of water at four different salinity levels.**

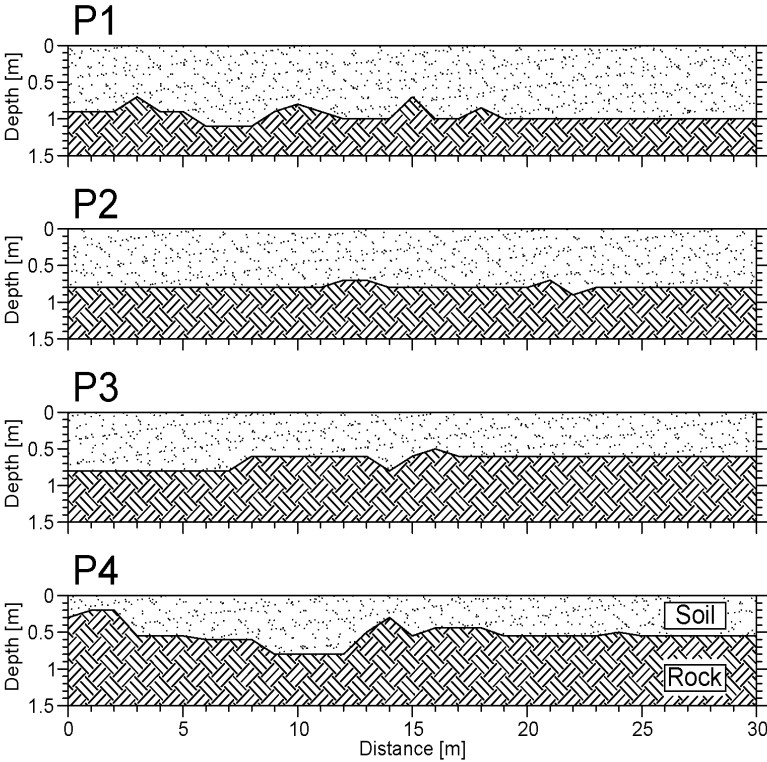

**Figure 2: Measured spatial distribution of the soil depth along four experimental plots, P1, P2, P3 and P4.**


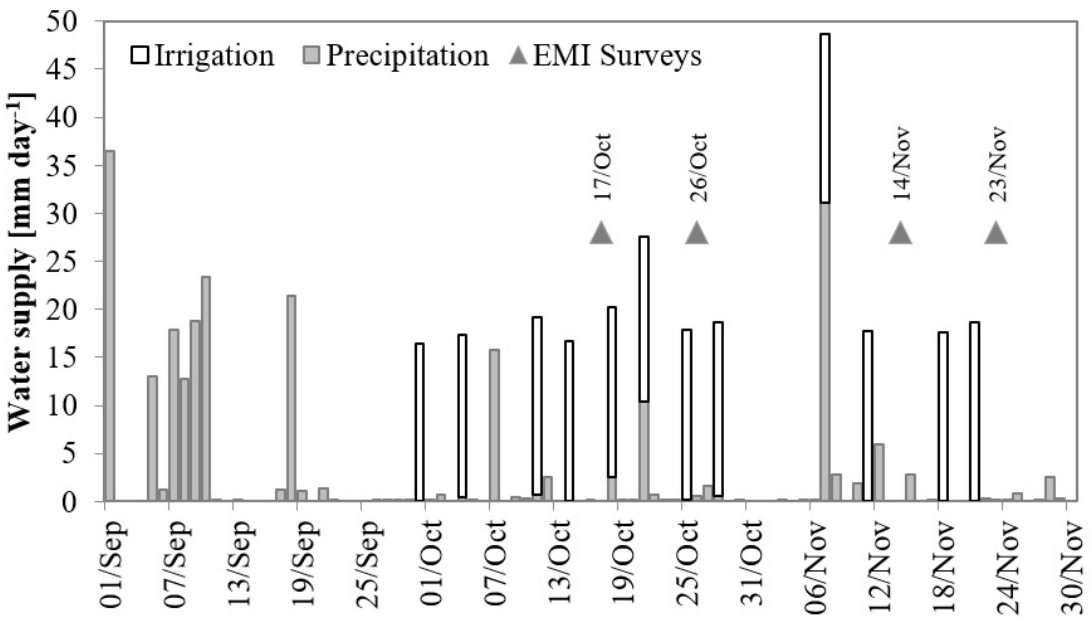

**Figure 3: The details of irrigation events and precipitation information during the experiment. The dates of EMI measurements are marked with triangular.**

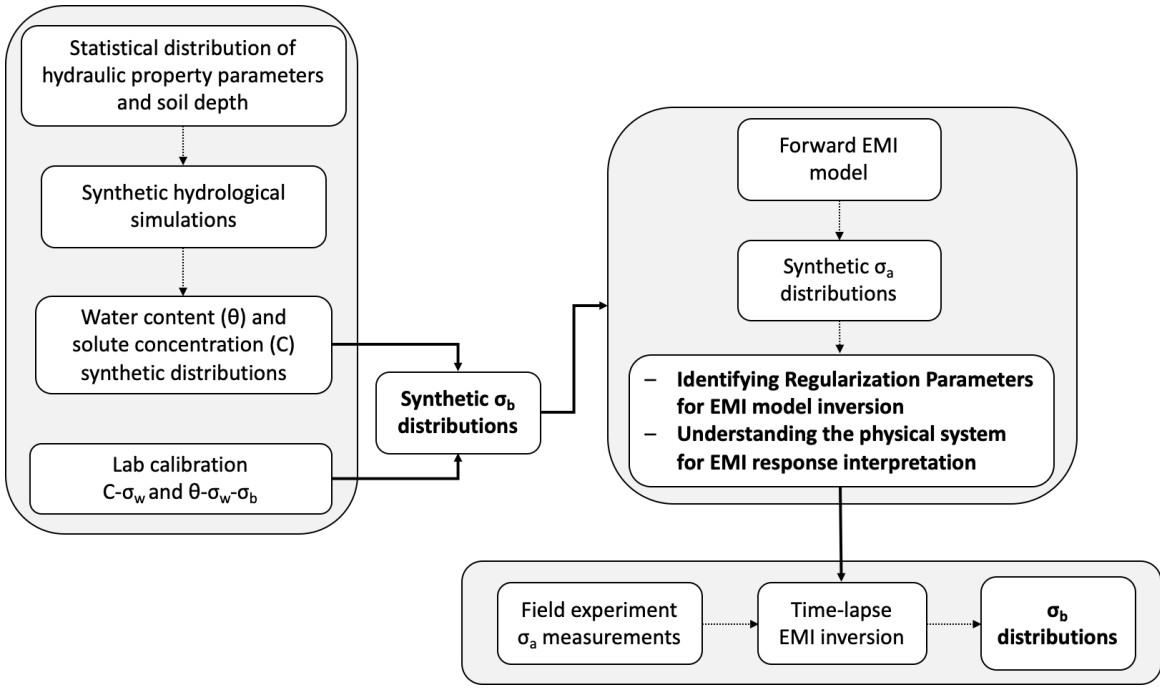

**Figure 4: Flow chart of the applied procedure with the key steps in bold. The procedure is explained in details in the text**


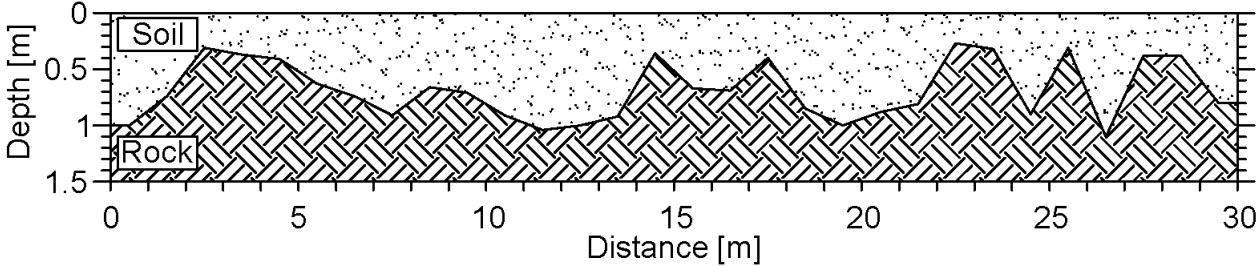

**Figure 5. Simulated spatial distribution of the soil depth for the selected scenario.**


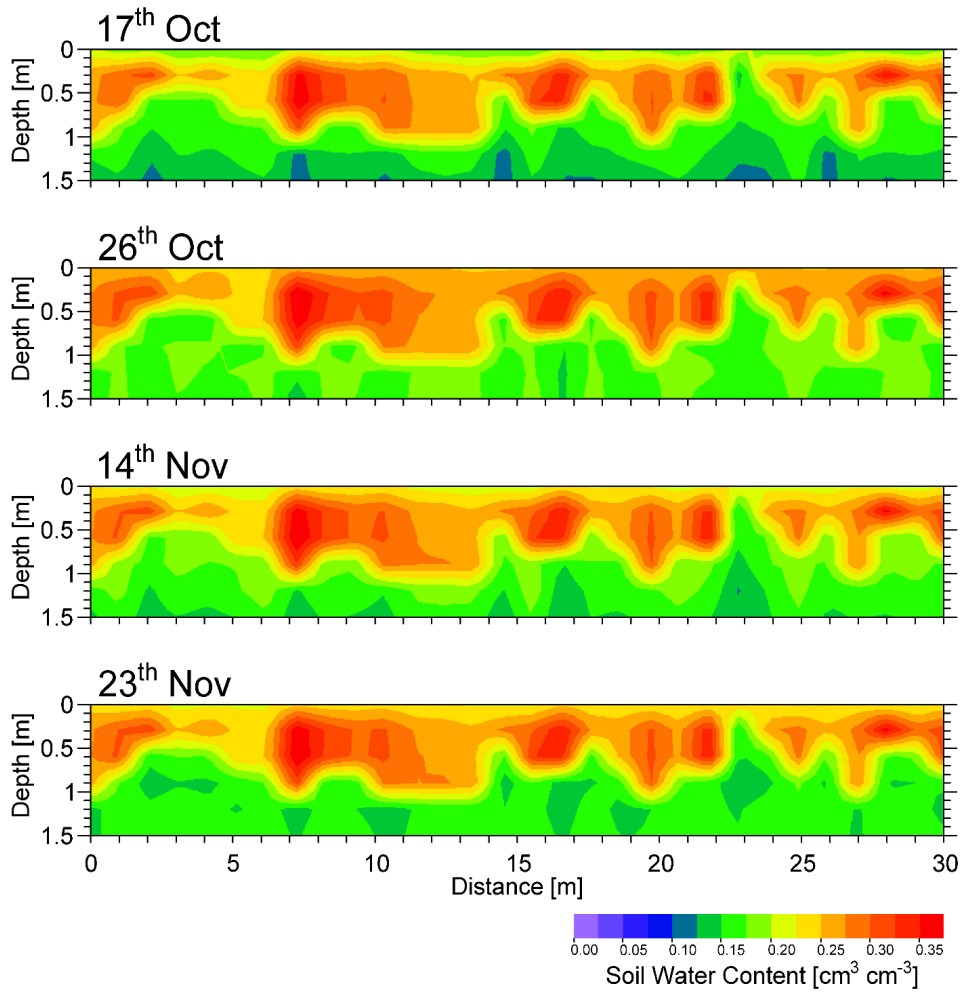


**Figure 6: Soil water content distribution simulations for the selected scenario: a) 17 Oct, b) 26 Oct, c) 14 Nov and d) 23 Nov 2016.**

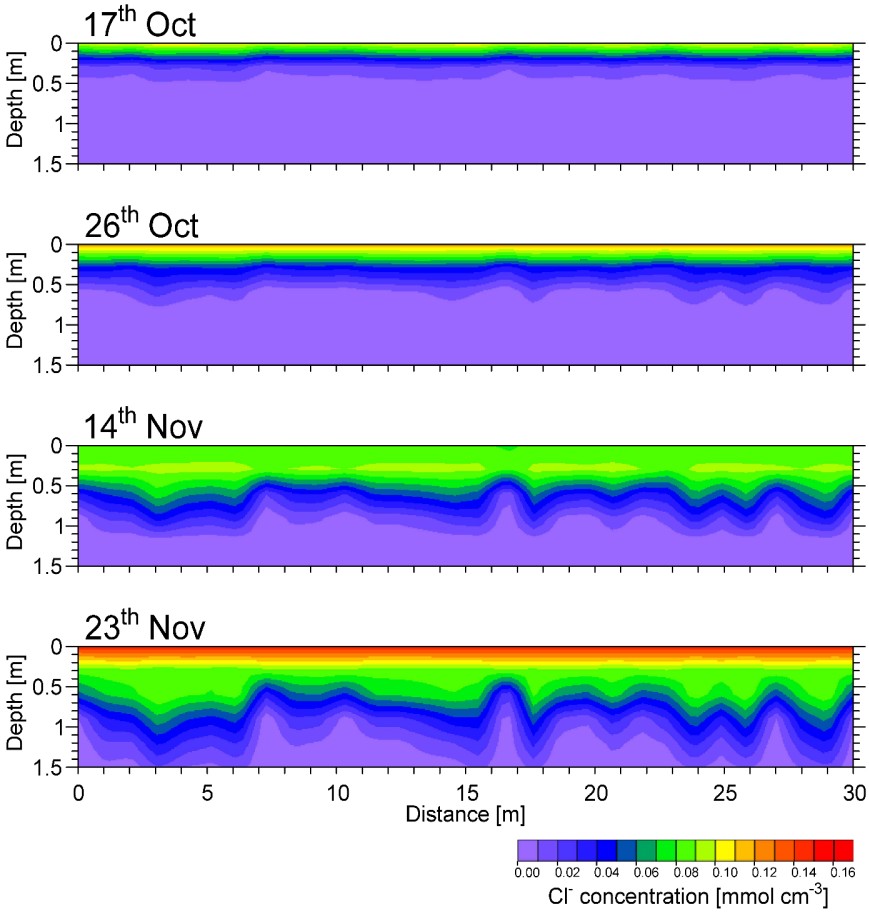

**Figure 7: Soil Cl⁻ concentration distributions simulations for the selected scenario: a) 17 Oct, b) 26 Oct, c) 14 Nov and d) 23 Nov 2016**


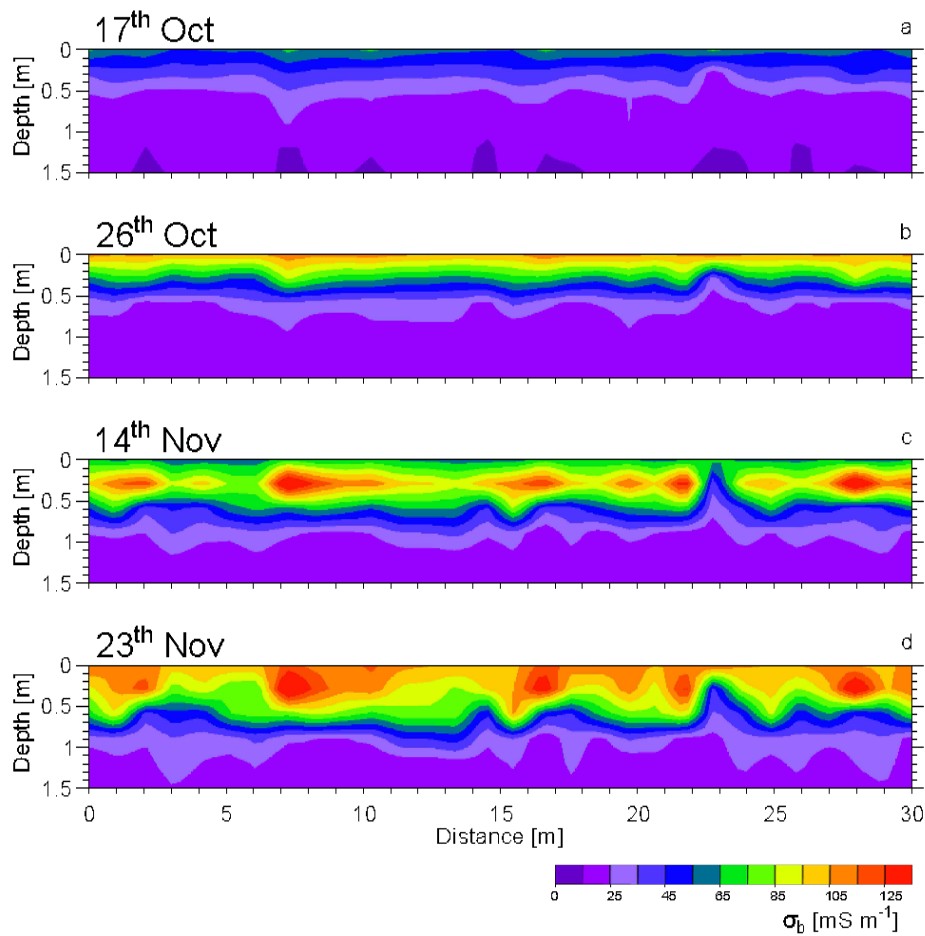

**Figure 8: $\sigma_b$ distributions simulations for the selected scenario: a) 17 Oct, b) 26 Oct, c) 14 Nov and d) 23 Nov 2016.**

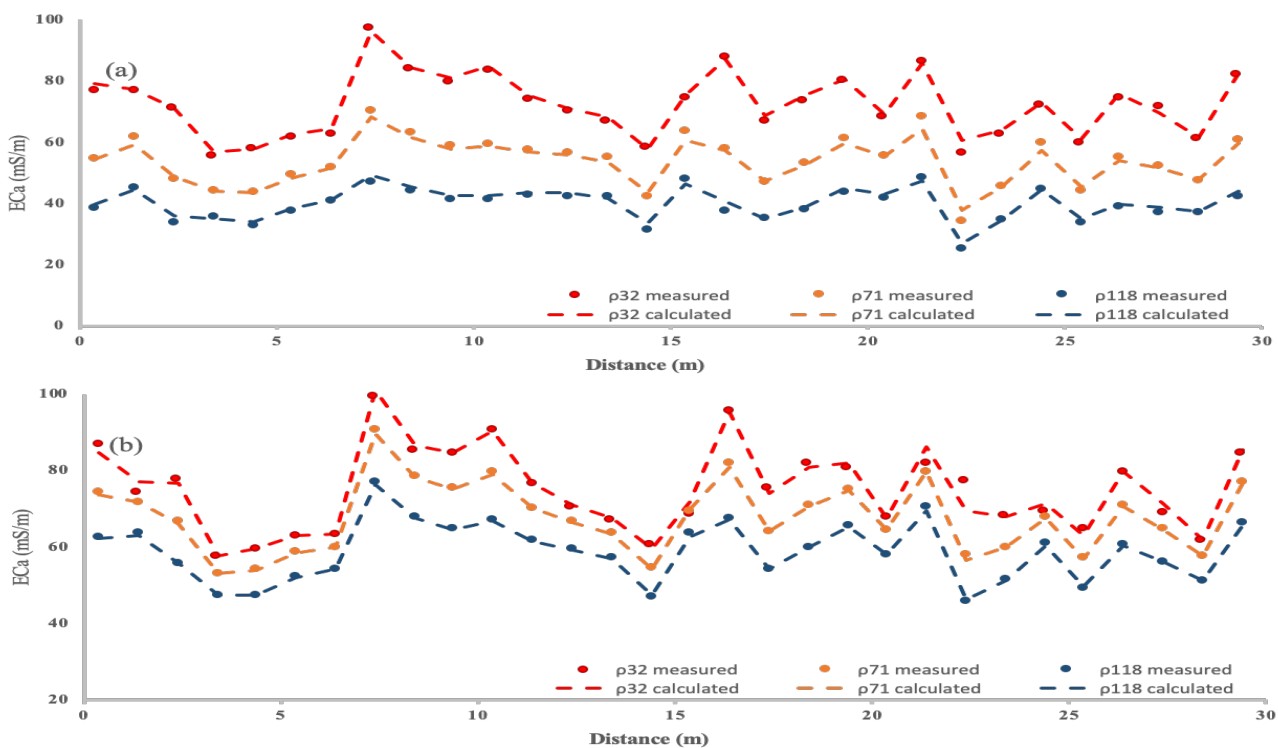

**Figure 9:** $\sigma_a$ data generated from the forward calculation of the $\sigma_b$ distributions on 23 Nov, shown in Fig. 8d. (a) HCP mode and (b) VCP mode configurations are displayed.


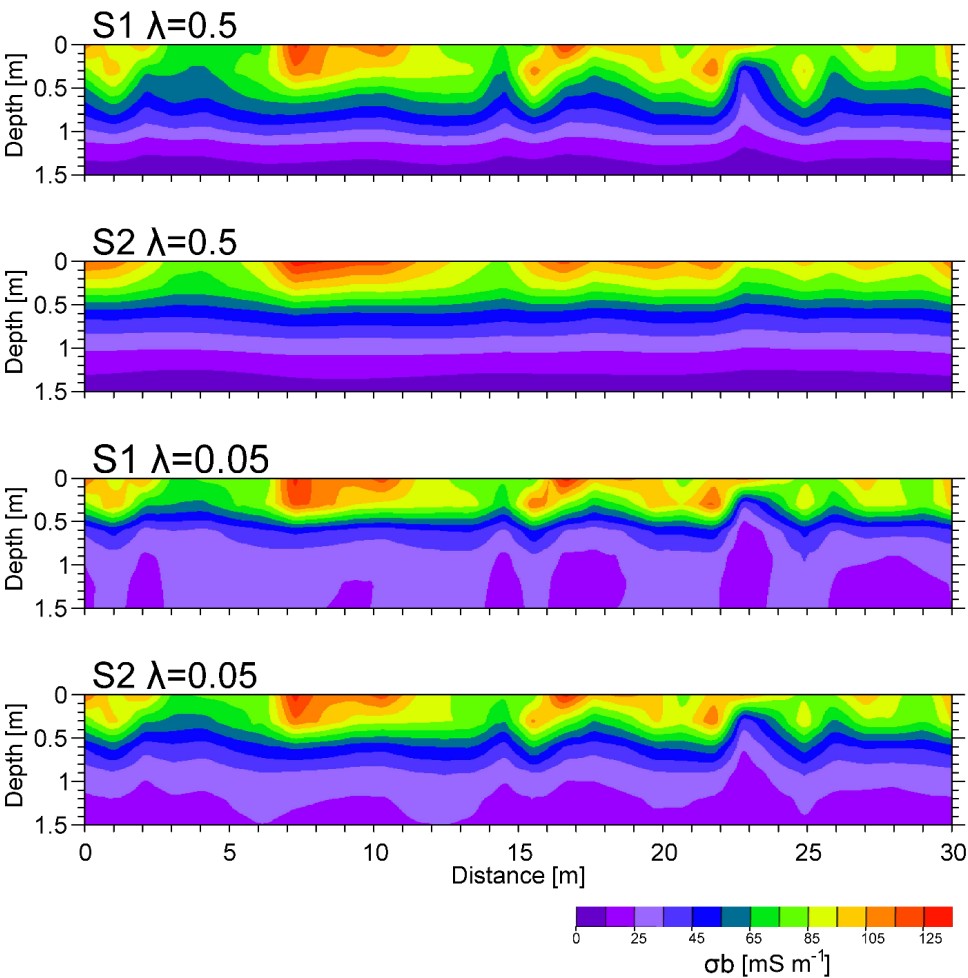

**Figure 10: $\sigma_b$ distributions obtained from the inversion of the synthetic $\sigma_a$ data (23 Nov), shown in Fig. 9, using four different combinations of two algorithms, S1 and S2, and $\lambda$ parameters.**

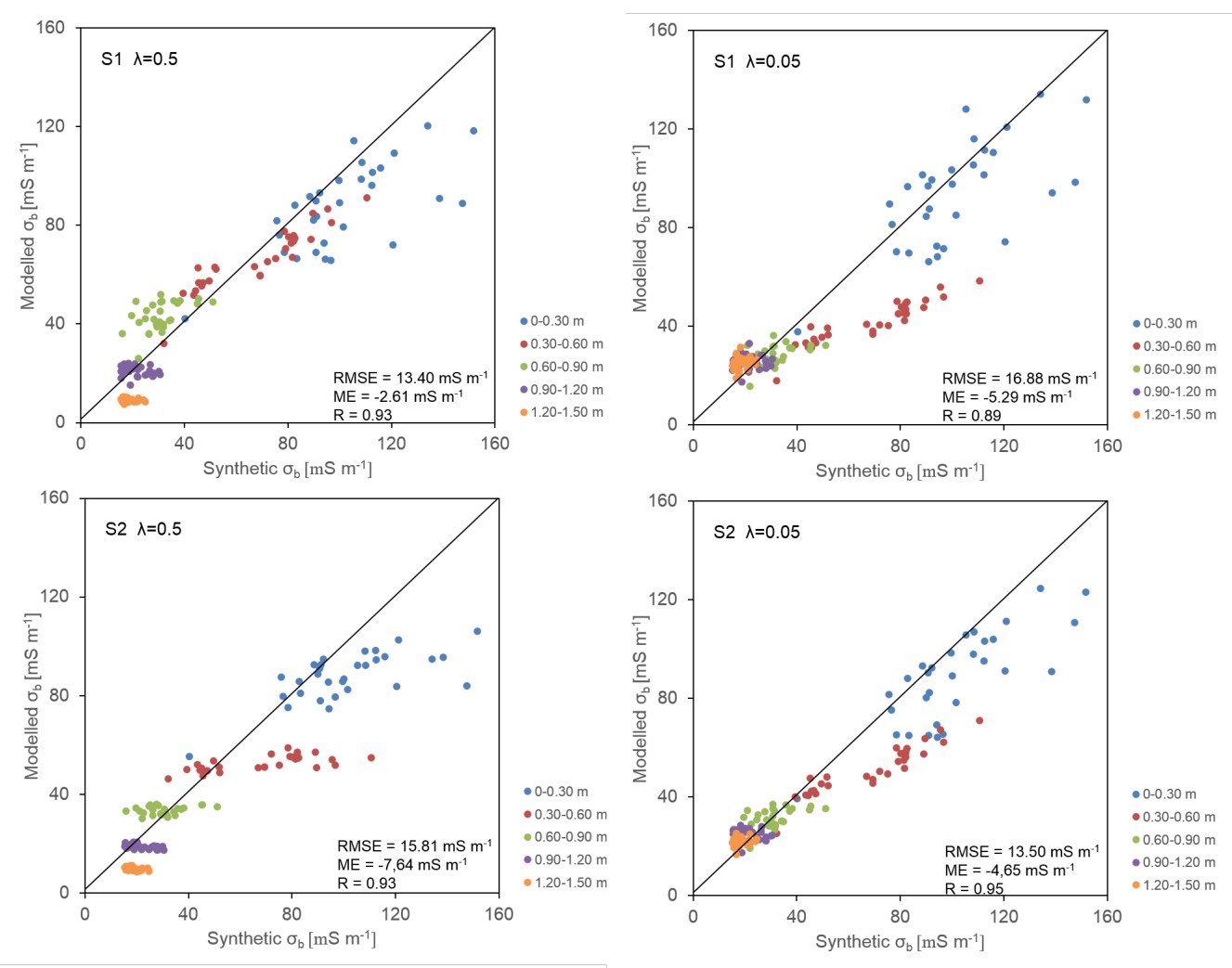

**Figure 11: Synthetic versus modelled σ<sub>b</sub> distribution for four different combinations of S and λ inversion parameters. Different colours refer to different depths. Main statistics are also reported.**


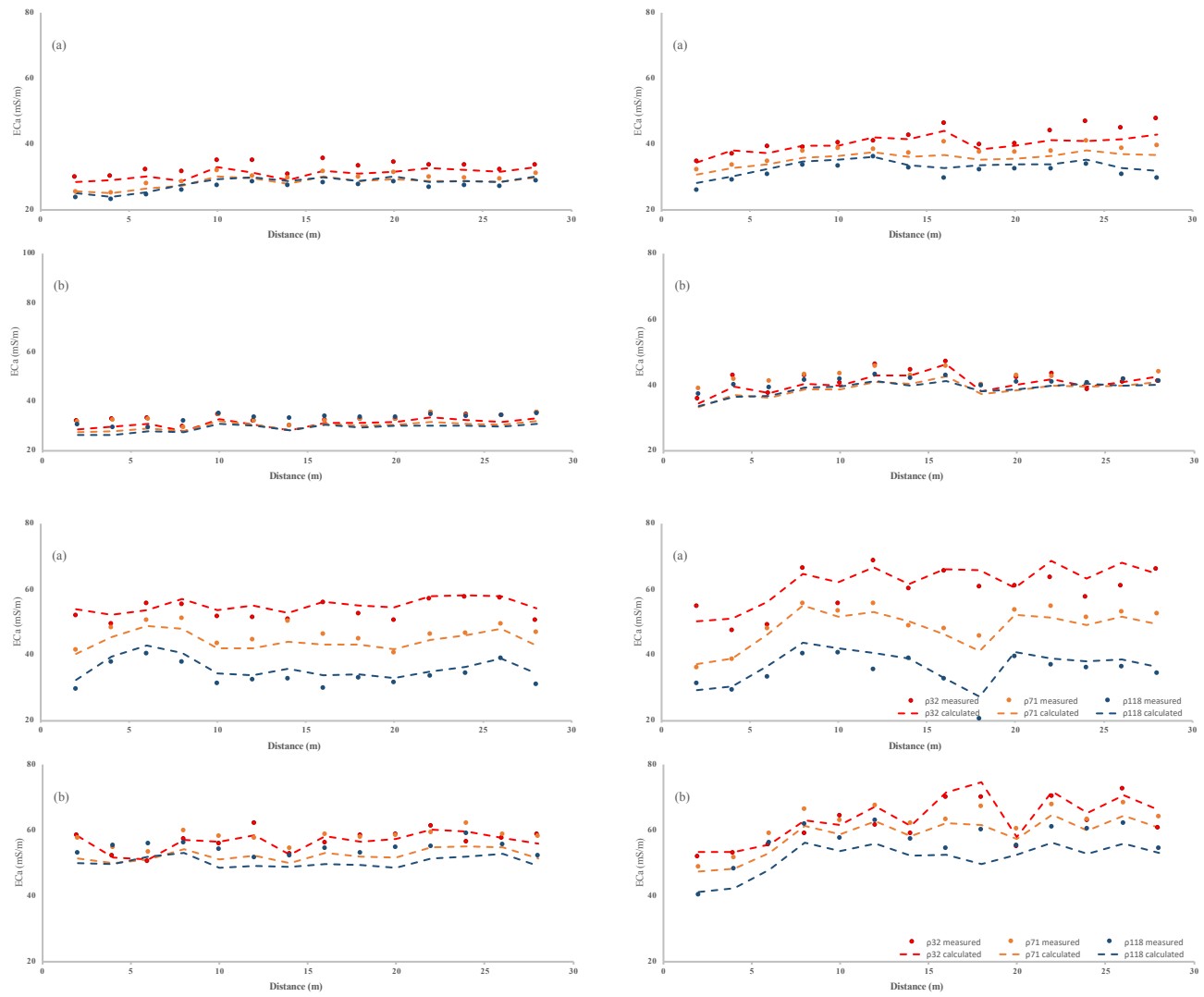

**Figure 12: σ$_a$ distribution of the middle transect in four experimental plots P1, P2, P3 and P4 referred to 23 Nov. (a) and (b)show VCP and HCP configuration, respectively.**

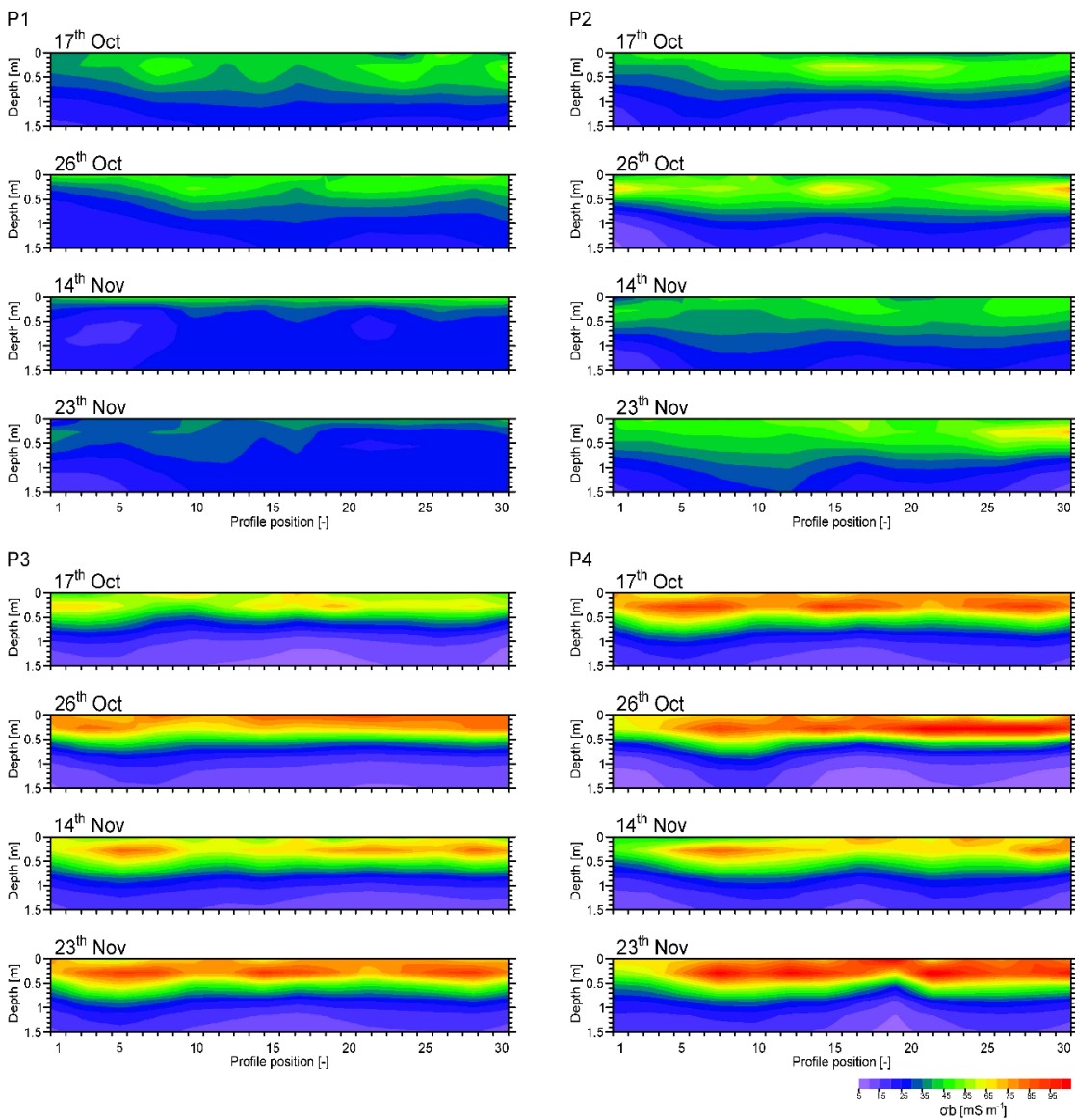

**Figure 13: Spatio-temporal distribution of $\sigma_b$ along the middle transect in four experimental plots P1, P2, P3 and P4 and for four selected dates: 17 Oct, 26 Oct, 14 Nov and 23 Nov 2016.**
