# Peer review of "Assessing the dynamics of soil salinity with time-lapse inversion of electromagnetic data guided by hydrological modelling"

_Hydrology and Earth System Sciences, 2020_

## Referee Comment (RC1) · Anonymous Referee #1 · 11 Dec 2020

This manuscript addresses the use of soil water and solute modelling (HYDRUS 2D) for finetuning the inversion of electromagnetic induction (EMI) data in the context of soil salinity studies. A suitable dataset is available, consisting of detailed soil data for the soil water modelling exercise and time-lapse EMI data measured along transects that were treated with irrigation water of different salinity levels. The proposed methods are sound and appropriate. The manuscript is well written. Below some suggestions are provided that might be of use for tightening the focus and improve the structure of the manuscript. The focus should be on the EMI inversion, which is the relevant and novel part, not on the hydrological simulations nor on the field experiment. I can therefore recommend "minor revisions".

General comments: 1. There has been a lot of talk about the use of hydrological modelling to optimize or constrain inversion of EMI data, but no clear framework to do so has been proposed so far. This manuscript contributes to the development of such a framework. Therefore, the topic is timely, relevant and novel, while also of interest to practitioners of inversion of electromagnetic induction data. 2. Overall, a rather qualitative approach is taken in this manuscript when it comes to interpretation of the results. This contrasts heavily with the strong quantitative approach taken to model soil water and solute transport and to invert the EMI data. Readers might expect o more quantitative evaluation of the results. 3. The focus of the manuscript should be tightened to make clear to the reader from the beginning what the authors want to achieve. As it stands, apparently more attention is dedicated to the soil water and solute transport modelling than to the inversion of the EMI data. This should be reverted by discussing first the chosen inversion approach and the details of the different parameters. From this analysis it should become clear why and how soil water and solute modelling can be used to optimize the inversion parameters. Also, the objectives need to be rewritten according to the chosen focus. The results and discussion should be reorganized accordingly. Redundant information (information that is not used further on or not relevant) on the field experiment and hydrological modelling should be omitted. 4. The reader should be informed why a synthetic study is necessary in this case. I can understand that a synthetic study can provide information for the inversion, beyond the specific conditions of the field experiments. The synthetic study should be clearly distinguished and justified within the structure of the manuscript. When going through the manuscript, the reader also wonders why not simulate also the field conditions for the dates on which the EMI surveys are performed so that the forward models can be compared with field-observed EMI measurements? The synthetic part could be a "proof of concept" while the analysis of the real-world field EMI measurements could be considered an application/demonstration.

Specific comments: L41 this should be "a few centimeters" L50 Define sigmab in L50 upon first use L79-86. Reformulate the objectives in order to tighten the focus of the

manuscript. The performance of a controlled irrigation experiment, sigmaa monitoring or numerical simulation with a hydrological model are not objectives here. These tasks are part of the methods to achieve the objectives. According to the title and the introduction, the main objective should be "Parameter optimization and/or constraining in time-lapse EMI inversion using soil water and solute modelling", and more specific objectives should be strictly related with this main objective. L87 After the introduction the electromagnetic inversion methods should be first explained. This is the important and novel part of the methods section. Once this is done it becomes clear what is needed throughout the remainder of the manuscript: sigmaa measurements in the controlled field experiment and hydrological simulations which can be explained in subsequent sections. L89-140 This section can be substantially shortened. All information that is not used further on should be omitted. L110 Should this be Ko instead of Ks? L143-173. This description is confusing. Too many details are given so that it becomes difficult to see the wood for the trees. All irrelevant information should be omitted. The manuscript is not about the hydrological simulations but about how this information can be used to improve inversion of EMI data. L180-186. Start the section with this information. This is the novel and relevant part for this manuscript. Maybe a flowchart can be used to explain better what is actually done. L243-251 This section is very difficult to follow for non-specialists. Please rewrite this section so that also less experienced readers can understand what is done and what the meaning is of the different parameters and inversion variants. L253-256 This is confusing. If only one 12 dS/m scenario is used hereinafter it is not necessary to introduce all the available information in the preceding sections. Also, if only data from 4 dates are used do not provide information on 6 dates in the preceding sections. L258 It is unclear why a simulated bedrock needs to be used here if 4 of them were measured in the field experiment (Fig 1b). I understand that this is done to obtain more variability in the soil depth in order to see how this propagates through the hydrological model and the inversion. It this is the case, please state this clearly. Do not consider the hydrological simulation as a separate task but relate it to the inversion. L258-288. Shorten this section. Discuss what the relevance

is of these patterns for optimizing inversion parameters. L266 Which profiles do you refer to here? L283-283 Avoid repeating information. L300 I assume that rho32 refers to one of the signals that the EMI sensor provides, but this should be clearly introduced and explained in the M&M section. L308-344. I would expect a more quantitative approach here. Statistical measures (e.g. correlation coefficient, RMSE, MAE,...) for the correspondence between the section shown in Fig.6d and those shown in Fig 8 can be calculated for different inversion parameters and plotted in a graph. The optimal combination of parameters should show the best statistics. Also, more sophisticated map comparison methods can be used. Or variograms could be used to compare the spatial structure of the obtained profiles. L346-381. Is there any information (soil data or simulated water and solute transport data) available to validate these profiles? How can you check whether these sigmab maps really represent salinity and not only soil water content? You could optimize the parameter set for each transect by producing first simulated soil water and solute patterns and using this information for forward modelling as done in the synthetic example. It is still unclear why a synthetic example is needed in this manuscript Why not applying directly the method to the 4 monitored profiles?

---

## Referee Comment (RC2) · Anonymous Referee #2 · 23 Dec 2020

In this study, the authors carried out a time-lapse EMI survey over four experimental plots irrigated with water at four different salinity levels for three months. They examined how well the time-lapse EMI measurements and a time-lapse inversion algorithm can be used to monitor soil salinity variability in space and time through performing simulation experiments and inversion processes.

The proposed methods are up to date, innovative, and a new addition to agriculture geophysics. It could be used in the field of precision agriculture. The manuscript is well written. I would recommend publishing the paper after "minor revision". I have a few comments:

1- Interpretation of the results should be more quantitative rather than qualitative 2- Although the manuscript is well written, it is lengthy and has many details that should be omitted. 3- Why not to use the real field EMI measurements instead of synthetic data. 4- I am wondering if you could do a correction for EM data before using it. I would recommend reading Beamish (2011) paper to correct the FDEM data that is measured under LIN-condition. FDEM – apparent conductivity is no longer in a linear relationship with true soil conductivity in highly conductive geomaterials such as soil saturated with saltwater, which is your case.

---

## Author Comment (AC2) · 6 Jan 2021

Dear Referee #2,

We sincerely appreciate and thank you for your constructive comments on our manuscript. We will revise the manuscript according to your suggestions and comments. Our answers are placed below each of your comments.

Sincerely,

Angelo Basile on behalf of all authors

Anonymous Referee #2

[Figure]

In this study, the authors carried out a time-lapse EMI survey over four experimental plots irrigated with water at four different salinity levels for three months. They examined how well the time-lapse EMI measurements and a time-lapse inversion algorithm can be used to monitor soil salinity variability in space and time through performing simulation experiments and inversion processes. The proposed methods are up to date, innovative, and a new addition to agriculture geophysics. It could be used in the field of precision agriculture. The manuscript is well written. I would recommend publishing the paper after "minor revision".

- We would like to thank Anonymous Referee #2 for evaluating our manuscript. We highly appreciate the overall positive comments.

I have a few comments:

1- Interpretation of the results should be more quantitative rather than qualitative

- We agree with the Referee #2. We will revise the manuscript adding some quantitative evaluation of the results. Specifically, because the key point of the manuscript is the aid of a synthetic test to the inversion procedure we will present field ECa data and also discuss the inversion results with more details.

2- Although the manuscript is well written, it is lengthy and has many details that should be omitted.

- We agree with the Referee comment. We will revise and shorten the manuscript significantly. Please, you could refer to the detailed answers to the same issues given to the Referee # 1.

3- Why not to use the real field EMI measurements instead of synthetic data.

- Dear Referee #2, as your question is the same one asked by Referee #1, please read the detailed answer we gave Referee# 1 to her/his comment #4 and L346-381.

4- I am wondering if you could do a correction for EM data before using it. I would

recommend reading Beamish (2011) paper to correct the FDEM data that is measured under LIN-condition highly conductive geomaterials such as soil saturated with saltwater, which is your case.

- Although in most cases EMI sensors use the LIN approximation to convert quadrature component (Q) to field ECa measurements (which is only valid at low EC values), the GF Instruments including CMD-mini explorer that we used in this study use a manufacturer calibration at sites of known subsurface EC (e.g. McLachlan et al. 2020) for conversion of Q to ECa. The manufacturer calibration aims to obtain a more representative ECa value in the field to overcome the use of LIN approximation. We also measured all data manually in the field at ground level to maintain the manufacturer calibration which was acquired at ground level and to avoid the impact of the sensor height in both ECa measurements and inversion results. It is also worth mentioning that although the top-soil is conductive due to irrigation as the Referee pointed out, the soil (conductive layer) is very shallow and the presence of shallow resistive bedrock contribute significantly to the cumulative EMI response and as results the field ECa values are not very high (always below 100 mS/m) even for the transect irrigated with water at 12 dS/m. This was also expected from the synthetic data shown in Figure 7. Therefore, the obtained Eca values are still within the range that the relationship between Q and EC is monotonic and inversion of only ECa data will yield reliable results.

McLachlan, P., Blanchy, G., & Binley, A. (2020). EMagPy: Open-source standalone software for processing, forward modeling and inversion of electromagnetic induction data. Computers & Geosciences. https://doi.org/10.1016/j.cageo.2020.104561

---

## Author Response (AR1)

Dear Editor,

We changed the manuscript according to your requests, Referees comments and our answers to the Public Discussion.

All the changes in the new version of the manuscript are in track-change mode. The only exception concerns the M&M section that required a deep re-assembly. After the "cut and paste" of the different sub-sections, the modifications were accepted. For specific changed in M&M section the track-change mode is effective.

We have uploaded two *pdf files. One clean and another in track-change mode. Unfortunately, the line numbers of the two version are not the same. I was not able to solve this problem. I hope this will not troublesome. I apologize for this inconvenience.

Hereafter, for your advantage, are reported in red color – below the answers we already gave during the Public Discussion – the specific and detailed answers to Reviewers comments.

Sincerely,

Angelo Basile on behalf of all authors

**Anonymous Referee #1**
This manuscript addresses the use of soil water and solute modelling (HYDRUS 2D) for fine tuning the inversion of electromagnetic induction (EMI) data in the context of soil salinity studies. A suitable dataset is available, consisting of detailed soil data for the soil water modelling exercise and time-lapse EMI data measured along transects that were treated with irrigation water of different salinity levels. The proposed methods are sound and appropriate. The manuscript is well written. Below some suggestions are provided that might be of use for tightening the focus and improve the structure of the manuscript. The focus should be on the EMI inversion, which is the relevant and novel part, not on the hydrological simulations nor on the field experiment. I can therefore recommend "minor revisions".

*We would like to thank Anonymous Referee #1 for evaluating our manuscript. We highly appreciate the overall positive comments. While we agree that the manuscript should be revised to better clarify the aim of this study and the focus on the EMI inversion as we suggested in our answers to the comment 3, we should also highlight that the simulations and synthetic test in this study was not only used to optimize the inversion parameters, but also aims to explore a realistic electrical conductivity structure and determine the amount of information that can be retrieved in terms of soil process in an ideal case from EMI surveys. Please see our detailed answer to the comment 4.*

General comments:
1. There has been a lot of talk about the use of hydrological modelling to optimize or constrain inversion of EMI data, but no clear framework to do so has been proposed so far. This manuscript contributes to the development of such a framework. Therefore, the topic is timely, relevant and novel, while also of interest to practitioners of inversion of electromagnetic induction data.

*Thanks for the positive comment.*

2. Overall, a rather qualitative approach is taken in this manuscript when it comes to interpretation of the results. This contrasts heavily with the strong quantitative approach taken to model soil water and solute transport and to invert the EMI data. Readers might expect o more quantitative evaluation of the results.

*Many thanks for your comment. We agree with you and, therefore, we will revise the manuscript adding some quantitative evaluation of the results. Specifically, because the key point of the manuscript is the aid of a synthetic test to the inversion procedure, we will present the field ECa data and will also discuss the inversion results with more details.*

*Specifically, we have added:*

- *The Figure 11 where the statistical parameters of the inversion procedure has been reported. The results reported in the Figure has been thoroughly commented below the figure 11.*
- *the Figure 12 with the $\sigma_a$ distributions. This Figure has been commented in the sub-section 3.5. Inversion of the real time-lapse $\sigma_a$ field data.*

3. The focus of the manuscript should be tightened to make clear to the reader from the beginning what the authors want to achieve. As it stands, apparently more attention is dedicated to the soil water and solute transport modelling than to the inversion of the EMI data. This should be reverted by discussing first the chosen inversion approach and the details of the different parameters. From this analysis it should become clear why and how soil water and solute modelling can be used to optimize the inversion parameters. Also, the objectives need to be rewritten according to the chosen focus. The results and discussion should be reorganized accordingly. Redundant information (information that is not used further on or not relevant) on the field experiment and hydrological modelling should be omitted.

*Thanks for the constructive comment. We agree with your suggestions and - in case of a positive answer by the Editor - we will revise the manuscript in this light. We suggest merging section 2 with Material and methods (section 3), to present only Material and methods section (section 2) with following subsections: 2.1 CMD Mini-Explorer Configuration, 2.2 Electromagnetic Forward Model, 2.3 Time-lapse inversion. The experimental set-up (2.4) and synthetic hydrological simulations (2.5) will be then discussed as suggested by the Referee.*

*Specifically,*

- *We have deleted redundant information on the field experiment (you can check them thanks to the tracking-mode version of the manuscript)*
- *The M&M Section was revised accordingly to your suggestions. Specifically, after introduce the experimental set-up (2.1) we reported the key Section on EMI analysis (2.2) followed by the Section describing the Synthetic experiment (2.3); then, we have reported the adjunctive Section 2.4 on the $\vartheta$-$\sigma_w$-$\sigma_b$ calibration. Finally, as requested in another comments, we have added a scheme of the procedure (2.5) at the end of M&M.*
- *Finally, we have shortened as much as possible the section related to hydrological modelling (2.3.2.)*

4. The reader should be informed why a synthetic study is necessary in this case. I can understand that a synthetic study can provide information for the inversion, beyond the specific conditions of the field experiments. The synthetic study should be clearly distinguished and justified within the structure of the manuscript. When going through the manuscript, the reader also wonders why not simulate also the field conditions for the dates on which the EMI surveys are performed so that the forward models can be compared with field-observed EMI measurements? The synthetic part could be a "proof of concept" while the analysis of the real-world field EMI measurements could be considered an application/demonstration.

*This comment deals with the key concept behind this manuscript and the why we didn't use the real transect data. Therefore, a detailed answer is needed. We want to explore a realistic electrical conductivty structure and determine the amount of information that can be retrieved in an ideal case where all the information is know - hence a synthetic model.*

*Specifically, EMI sensors are more and more used for monitoring water content and solute concentrations in soil profiles, along transects and even in 3D space. The problem is that interpretation of EMI readings (ECa) is not an easy task, as it is difficult to understand the real vertical distribution of bulk electrical conductivity from ECa readings which integrate the whole soil profile response. Thus, having some preliminary idea of the physical system to be monitored by the EMI sensor is highly desirable.*

*As an alternative preliminary analysis carrying out hydrological simulations by applying real boundary conditions measured during an EMI sensor monitoring campaign. In any case, this would require the knowledge of the hydraulic properties in each point along the transect or, more in general, in the field to be monitored by the EMI. One can immediately realize that this is quite utopian, especially when the area to be monitored is relatively large (as previously recalled in the case of EMI measurements). By contrast, it is more common that, for a given area, one has the available statistical distribution of hydraulic properties. It is our case where statistical distribution is available for an area of about 2 ha where the transect is included. The statistical distribution of the hydraulic properties may thus be used for generating synthetic equiprobable realizations of the physical conditions the EMI sensor will experience during the monitoring.*

*This may be potentially crucial for addressing the inversion of EMI reading for at least two reasons:*

1. *By recalling that the EMI inversion is an ill-posed problem, the synthetic tests may be used to optimize the smoothing parameter to be used in the inversion of EMI data. It is known that this is a crucial task in the inversion of EMI data. In practice, by running several hydrological simulations on as many synthetic transects, one obtains several potential distributions of $\sigma_b$, coming from the combination of water contents and concentrations obtained by the simulations. These distributions of $\sigma_b$ are potential, but realistic because they are based on real boundary conditions and real soil hydraulic properties variability Then, the $\sigma_b$ distributions, may be, in turn, used in a forward approach to rebuild the ECa readings the EMI sensor would have seen in those transects. This way, the smoothing parameter may be found as the value where most of the inversions converge (this is just what we did).*

2. *The synthetic transects may be used for preliminary understanding of how the physical context may influence the EMI readings (if for example, the EMI readings are more sensitive to the pedological context (i.e. layers with very contrasting characteristics depth to bedrock, resistive clay layers, stones, etc.) or to the horizontal and vertical variability of the hydraulic properties.*

*These concepts were discussed both in the Introduction section and in the Material and Methods section. In case of a positive answer by the Editor, we will make them clearer, by slightly revising the manuscript.*

*We have added a discussion in the last part of the Introduction section, highlighting the concepts above reported, by specifically stressing the why a synthetic text is required and consequently better refining the objectives.*

Specific comments:
L41 this should be "a few centimeters"

*Thanks. We will change it.*

*Done*

L50 Define sigmab in L50 upon first use

*We defined $\sigma_b$ in L39.*

L79-86. Reformulate the objectives in order to tighten the focus of the manuscript. The performance of a controlled irrigation experiment, sigmaa monitoring or numerical simulation with a hydrological model are not objectives here. These tasks are part of the methods to achieve the objectives. According to the title and the introduction, the main objective should be "Parameter optimization and/or constraining in time-lapse EMI inversion using soil water and solute modelling", and more specific objectives should be strictly related with this main objective.

*Thanks for the comment. We agree with you that the objectives were not properly formulated. Subject to a positive answer by the Editor, we will revise this paragraph according to your suggestions.*

*Specifically, the main objective was changed as follows:*
*The main objective of this paper is to propose an approach to improve the optimization of parameters and constrains in time-lapse EMI inversion using soil water and solute modelling. In the paper we will show how the synthetic tests may be used to guide the optimization of inversion parameters and understand the impact of solute concentration and water content variations on EMI $\sigma_a$ readings.*

L87 After the introduction the electromagnetic inversion methods should be first explained. This is the important and novel part of the methods section. Once this is done it becomes clear what is needed throughout the remainder of the manuscript: sigmaa measurements in the controlled field experiment and hydrological simulations which can be explained in subsequent sections.

*We agree with you. We will thoroughly revise sections 2 and 3. In the revised version, section 2 (Material and methods) will start with 2.1 CMD Mini-Explorer Configuration, 2.2 Electromagnetic Forward Model, and then 2.3 Time-lapse inversion. The experimental set-up (2.4) and synthetic hydrological simulations (2.5) will be then discussed as suggested by the Referee, according also to the general comment #3.*

*As already mentioned in our answer to your question 3, the M&M Section was revised. Specifically, after introduce the experimental set-up (2.1) we reported the key Section on EMI analysis (2.2) followed by the Section describing the Synthetic experiment (2.3); then, we have reported the adjunctive Section 2.4 on the $\vartheta$-$\sigma_w$-$\sigma_b$ calibration. Finally, as requested in another comments, we have added a scheme of the procedure (2.5) at the end of M&M.*

L.89140 This section can be substantially shortened. All information that is not used further on should be omitted.

*This section, "background information" has been omitted and relevant information were included into the new Section 2.3.*

L110 Should this be Ko instead of Ks?

*Thanks. We have changed it.*

L143-173. This description is confusing. Too many details are given so that it becomes difficult to see the wood for the trees. All irrelevant information should be omitted. The manuscript is not about the hydrological simulations but about how this information can be used to improve inversion of EMI data.

*We will revise thoroughly sections 2 and 3, as previously stated to make the M&M easier to follow. Concerning this part on modelling procedure, we partially disagree with the comment of the Referee. As reported in our answer to comment #4 the simulation runs were performed by using actual boundary conditions both for water and solute balance, actual input (i.e. rain, irrigation and salt), etc. We used the statistical distribution of the hydraulic properties. We believe that the reader should be aware of the procedure we applied to generate the hydraulic properties for the synthetic test. However, we will shorten this part, deleting the redundant and not strictly required information.*

*See previous answer on M&M Section re-assembly, general comment #3.*

L180-186. Start the section with this information. This is the novel and relevant part for this manuscript. Maybe a flowchart can be used to explain better what is actually done.

*We agree with your suggestions. We will move this part at the beginning of the section and we will insert a flowchart.*

*A specific sub-section (2.5) reporting a flow chart of the proposed procedure has been added at the end of M&M.*

L243-251 This section is very difficult to follow for non-specialists. Please rewrite this section so that also less experienced readers can understand what is done and what the meaning is of the different parameters and inversion variants.

*We agree with your suggestion. We will revise the section to make it easier for less experienced readers to understand.*

*A new part to help non-specialist in the paper reading was added in the sub-section 2.2.3. Time-lapse inversion.*

L253-256 This is confusing. If only one 12 dS/m scenario is used hereinafter it is not necessary to introduce all the available information in the preceding sections. Also, if only data from 4 dates are used do not provide information on 6 dates in the preceding sections.

*We will remove this paragraph. We will also revise the manuscript in this regard and only provide information on 4 dates of measurements in the preceding sections.*

*Done. See tracked modifications.*

L258 It is unclear why a simulated bedrock needs to be used here if 4 of them were measured in the field experiment (Fig 1b). I understand that this is done to obtain more variability in the soil depth in order to see how this propagates through the hydrological model and the inversion. If this is the case, please state this clearly. Do not consider the hydrological simulation as a separate task but relate it to the inversion.

*Thanks for your comment. The answer is yes, we will clearly write it. The depth of the bedrock is a crucial parameter for the water flows, therefore we have used its statistical distribution (measured in 40 points in the field) as well. Please, see also our detailed answer to the general comment #4.*

*Nothing to add.*

L258-288. Shorten this section. Discuss what the relevance is of these patterns for optimizing inversion parameters.

*The simulations in this study was used not only for the optimization of the EMI inversion parameters, but also for a better understanding of the complexity of the spatial and temporal variability of $\sigma_b$ due to simultaneous*

*variations of solute concentration and water content in irrigated lands as discussed in the abstract (L 21) and introduction (L 75) of the paper. This section provides valuable information for the interpretation of EMI models and was frequently addressed in section 4.5 for interpretation of spatial and temporal variability of $\sigma_b$ distribution in terms of soil process, observed in the simulations. See also answer to comment #4.*

*Nothing to add.*

L266 Which profiles do you refer to here?

*The soil depth shown in Figure 3.*

L283-283 Avoid repeating information.

*Thanks. We will delete repeated information (e.g. the value of dispersivity) here and elsewhere in the manuscript.*

*Done.*

L300 I assume that rho32 refers to one of the signals that the EMI sensor provides, but this should be clearly introduced and explained in the M&M section.

*This information can be found in section 2.1 (old version) and in the Section 2.2.1. "Characteristics of the EMI sensor" of the new manuscript version*

L308-344. I would expect a more quantitative approach here. Statistical measures (e.g. correlation coefficient, RMSE, MAE, . . .) for the correspondence between the section shown in Fig.6d and those shown in Fig 8 can be calculated for different inversion parameters and plotted in a graph. The optimal combination of parameters should show the best statistics. Also, more sophisticated map comparison methods can be used. Or variograms could be used to compare the spatial structure of the obtained profiles.

*We agree with your suggestion. We did the statistical analysis and we will provide it in the revised version, subject to a positive answer by the Editor.*

See answer to the general comment #2

L346-381. Is there any information (soil data or simulated water and solute transport data) available to validate these profiles? How can you check whether these sigmab maps really represent salinity and not only soil water content? You could optimize the parameter set for each transect by producing first simulated soil water and solute patterns and using this information for forward modelling as done in the synthetic example. It is still unclear why a synthetic example is needed in this manuscript Why not applying directly the method to the 4 monitored profiles?

*As mentioned in our detailed answer to the general comment #4, the actual distribution of the soil hydraulic parameters along the transects are not available. Therefore, a direct comparison between variables estimated by simulations and by EMI measurements is unfeasible.*

*As reported above, to prevent misunderstanding on this key issue a better explanation on the introduction section was provided.*
* * *
**Anonymous Referee #2**

In this study, the authors carried out a time-lapse EMI survey over four experimental plots irrigated with water at four different salinity levels for three months. They examined how well the time-lapse EMI measurements and a time-lapse inversion algorithm can be used to monitor soil salinity variability in space and time through performing simulation experiments and inversion processes. The proposed methods are up to date, innovative, and a new addition to agriculture geophysics. It could be used in the field of precision agriculture. The manuscript is well written. I would recommend publishing the paper after "minor revision".

*We would like to thank Anonymous Referee #2 for evaluating our manuscript. We highly appreciate the overall positive comments.*

I have a few comments:
1- Interpretation of the results should be more quantitative rather than qualitative

*We agree with the Referee #2. We will revise the manuscript adding some quantitative evaluation of the results. Specifically, because the key point of the manuscript is the aid of a synthetic test to the inversion procedure we will present field ECa data and also discuss the inversion results with more details.*
See detailed answer to Referee #1 on this issue. Specifically, the answer to his/her comment #2.

2- Although the manuscript is well written, it is lengthy and has many details that should be omitted.

*We agree with the Referee comment. We will revise and shorten the manuscript significantly. Please, you could refer to the detailed answers to the same issues given to the Referee # 1.*

3- Why not to use the real field EMI measurements instead of synthetic data.

*Dear Referee #2, as your question is the same one asked by Referee #1, please read the detailed answer we gave Referee# 1 to her/his comment #4 and L346-381.*

4- I am wondering if you could do a correction for EM data before using it. I would recommend reading Beamish (2011) paper to correct the FDEM data that is measured under LIN-condition highly conductive geomaterials such as soil saturated with saltwater, which is your case.

*Although in most cases EMI sensors use the LIN approximation to convert quadrature component (Q) to field $\sigma_a$ measurements (which is only valid at low EC values), the GF Instruments including CMD-mini explorer that we used in this study use a manufacturer calibration at sites of known subsurface EC (e.g. McLachlan et al. 2020) for conversion of Q to $\sigma_a$. The manufacturer calibration aims to obtain a more representative $\sigma_a$ value in the field to overcome the use of LIN approximation. We also measured all data manually in the field at ground level to maintain the manufacturer calibration which was acquired at ground level and to avoid the impact of the sensor height in both $\sigma_a$ measurements and inversion results. It is also worth mentioning that although the top-soil is conductive due to irrigation as the Referee pointed out, the soil (conductive layer) is very shallow and the presence of shallow resistive bedrock contribute significantly to the cumulative EMI response and as results the field ECa values are not very high (always below 100 mS/m) even for the transect irrigated with water at 12 dS/m. This was also expected from the synthetic data shown in Figure 7. Therefore, the obtained $\sigma_a$ values are still within the range that the relationship between Q and EC is monotonic and inversion of only $\sigma_a$ data will yield reliable results.*

*McLachlan, P., Blanchy, G., & Binley, A. (2020). EMagPy: Open-source standalone software for processing, forward modeling and inversion of electromagnetic induction data. Computers & Geosciences. https://doi.org/10.1016/j.cageo.2020.104561*